# OmniVL-Guard: Towards Unified Vision-Language Forgery Detection and Grounding via Balanced RL

Jinjie Shen [* 1 2 3]    Jing Wu [* 1 3]    Yaxiong Wang [† 1 3]    Lechao Cheng [1 3]    Shengeng Tang [1]
Tianrui Hui [1 3]    Nan Pu [1 3]    Zhun Zhong [1 3]

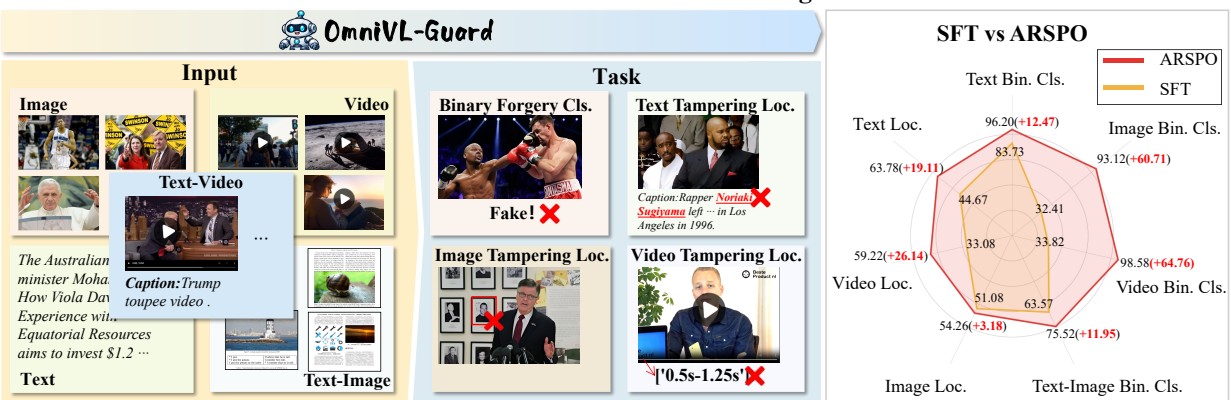

*Figure 1.* This work explores the task of omnibus vision-language forgery detection and grounding (left). In this unified setting, simple Supervised Fine-Tuning (SFT) can not achieve coordinated performance improvements. In response, we propose Adaptive Reward Scaling Policy Optimization (ARSPO), achieving balanced optimization in detecting and grounding tasks (right).

## Abstract

Existing forgery detection methods are often limited to uni-modal or bi-modal settings, failing to handle the interleaved text, images, and videos prevalent in real-world misinformation. To bridge this gap, this paper targets to develop a unified framework for omnibus vision-language forgery detection and grounding. In this unified setting, the interplay between diverse modalities and the dual requirements of simultaneous detection and localization pose a critical "difficulty bias" problem: the simpler veracity classification task tends to dominate the gradients, leading to suboptimal performance in fine-grained grounding during multi-task optimization. To address this challenge, we propose **OmniVL-Guard**, a balanced reinforcement learning framework for omnibus vision-language forgery detection and ground-

ing. Particularly, OmniVL-Guard comprises two core designs: Self-Evolving CoT Generation and Adaptive Reward Scaling Policy Optimization (ARSPO). Self-Evolving CoT Generation synthesizes high-quality reasoning paths, effectively overcoming the cold-start challenge. Building upon this, Adaptive Reward Scaling Policy Optimization (ARSPO) dynamically modulates reward scales and task weights, ensuring a balanced joint optimization. Extensive experiments demonstrate that OmniVL-Guard significantly outperforms state-of-the-art methods and exhibits zero-shot robust generalization across out-of-domain scenarios. The dataset and code are publicly available at https://github.com/shen8424/OmniVL-Guard.

## 1. Introduction

The rapid evolution of generative AI has significantly lowered the barrier for fabricating convincing misinformation, posing severe risks to multimedia information security (Haydarov et al., 2024; Li et al., 2024b; Abdelnabi et al., 2022). Consequently, the problem of forgery detecting and manipulation grounding for multi-modal media, aiming to verify veracity and localize manipulated regions, have garnered

*Equal contribution †Corresponding author. [1]School of Computer Science and Information Engineering, Hefei University of Technology, Hefei, China [2]Wuhan University, Wuhan, China [3]Lab for Intelligence and visiON (LION). First author: Jinjie Shen <shenjinjie22@gmail.com>. Corresponding to Yaxiong Wang <wangyx@hfut.edu.cn>.
*Proceedings of the 43rd International Conference on Machine Learning*, Seoul, South Korea. PMLR 306, 2026. Copyright 2026 by the author(s).

*Table 1.* Motivation Analysis: SOTA MLLMs Limitations and RL Optimization Imbalance.

*(a)* Performance of SOTA MLLMs across Forgery Detection Tasks. Notably, these models exhibit significant performance bottlenecks in the three localization tasks. Consequently, directly utilizing them to generate CoT data is inefficient due to frequent hallucinations and errors.

| Method | Bin. Cls. | Image Loc. | Text Loc. | Video Loc. |
|---|---|---|---|---|
| GPT5 | 73.71 | 19.01 | 34.96 | 24.60 |
| Gemini3 | 74.54 | 21.32 | 32.77 | 24.11 |
| Seed1.6 | 72.99 | 20.97 | 31.86 | 26.29 |

*(b)* Comparison of optimization balance in multi-task scenarios under $FSFR_{sft}$ and $FSFR_{rl}$. Our ARSPO achieves balanced improvements across all tasks.

| Method | Bin. Cls. | Image Loc. | Text Loc. | Video Loc. |
|---|---|---|---|---|
| SFT | 53.38 | 51.08 | 44.67 | 33.08 |
| + GRPO | 89.42 (+36.0) | 50.96 (-0.1) | 55.31 (+10.6) | 42.79 (+9.7) |
| + SAPO | 90.75 (+37.4) | 51.24 (+0.2) | 54.33 (+9.7) | 44.10 (+11.0) |
| **+ ARSPO** | **90.85** (+37.5) | **54.26** (+3.2) | **63.78** (+19.1) | **59.22** (+26.1) |

substantial attention (Zhang et al., 2025b; Yuchen et al., 2026; Jingchun et al., 2026; Tang et al., 2025a). However, existing paradigms (Shen et al., 2025; Wen et al., 2025; Tang et al., 2021) are predominantly confined to specific uni-modal or bi-modal settings (e.g., sole image or Image-Text). This restriction sharply contrasts with modern social media platforms, where misinformation is inherently diverse modalities, characterized by the simultaneous coexistence of text, images, and videos (Tang et al., 2025b; Jiang et al., 2025). As a result, current methods are incapable of processing these diverse modalities in an integrative manner.

To bridge this gap, we focus to develop a unified framework capable of processing the dominant vision-language modalities in social media—text, images, and videos—within a single paradigm. However, this undertaking is intrinsically challenging, as it requires the model to jointly perform veracity classification and fine-grained manipulation grounding. The difficulty is further exacerbated in omni-modal setting, where the model must reason across heterogeneous data types simultaneously. Standard Supervised Fine-Tuning (SFT) fails to yield consistent gains across modalities due to insufficient reasoning capabilities (Figure 1). In contrast, Reinforcement Learning (RL) has proven effective in endowing Multi-modal Large Language Models (MLLMs) with robust generalization and reasoning capabilities (DeepSeek-AI, 2025). Consequently, we pioneer the integration of RL into the domain of multi-modal Forgery detection and grounding, aiming to leverage the exploration mechanisms of RL to guide the model toward superior capabilities.

Following the paradigm of DeepSeek-R1 (DeepSeek-AI, 2025), implementing RL for forgery detection and grounding requires high-quality Chain-of-Thought (CoT) data for cold-start to facilitate complex reasoning. However, as demonstrated in Table 1a, general-purpose large models often lack the fine-grained perception required to identify subtle manipulations. This limitation makes it difficult to generate effective reasoning paths for complex forensic tasks. To overcome this, we introduce a Self-Evolving CoT Generation strategy that iteratively refines the reasoning process to produce high-quality CoT. We first collect source data from diverse public datasets and split it into two subsets

for SFT cold-start and RL optimization, respectively. To construct high-quality CoT annotations for SFT, we distill a small seed CoT subset from the SFT data and apply a self-evolving procedure to iteratively expand it, resulting in a high-quality SFT dataset. This dataset is then combined with the RL data splits to form our Full-Spectrum Forensic Reasoning (FSFR) dataset.

FSFR facilitates effective SFT cold-start and RL optimization for the OmniVL forgery detection and grounding tasks. While conventional RL algorithms, such as GRPO (DeepSeek-AI, 2025), demonstrate stronger optimization capability, the training dynamics are often dominated by the easier classification objective in our practice, as shown in Table 1b. This imbalance results in insufficient optimization of the more challenging grounding task. For example, although GRPO boosts classification accuracy by 36%, the performance on image grounding deteriorates. This imbalance underscores the need for a more effective optimization strategy for multi-task learning in omnibus vision-language modal forgery detection and grounding.

To tackle the aforementioned issue, we propose ARSPO (**A**daptive **R**eward **S**haping **P**olicy **O**ptimization) to facilitate balanced multi-task reinforcement learning. Through a mathematical decomposition of the optimization gradient, we analyze the impact of task difficulty and identify that the reward mapping function directly governs gradient transfer efficiency. Guided by this analysis, we design specialized reward functions that incentivize the model to tackle high-difficulty tasks. Furthermore, we introduce a dynamic coefficient adjustment strategy that adaptively modulates task weights based on real-time learning states to ensure an optimal equilibrium. Extensive experiments demonstrate that ARSPO not only achieves superior in-domain performance but also exhibits robust zero-shot generalization in out-of-domain scenarios.

Integrating Self-Evolving CoT Generation with ARSPO, we finally develop our **OmniVL-Guard**, a balanced RL framework for omnibus vision-language forgery detection and grounding. Overall, the main contributions are summarized as follows:

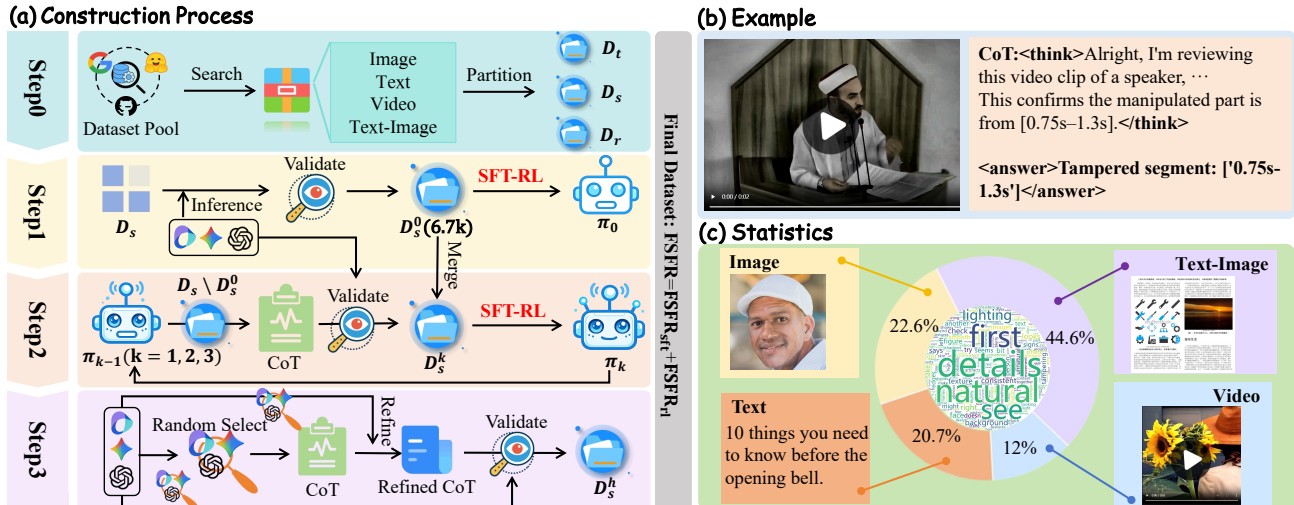

**Figure 2.** (a) The construction process of FSFR$_{sft}$. (b) An example from FSFR$_{sft}$. (c) Statistics of the union of FSFR$_{sft}$ and FSFR$_{rl}$, including the distribution and word clouds.

(1) **Unified Framework for omnibus vision-language forgery:** We explore and propose a unified VLM framework capable of processing images, videos, and text for forgery detection and grounding, while pioneering the integration of Reinforcement Learning to enhance forensic reasoning and cross-modal generalization.

(2) **Dataset FSFR:** We present FSFR, a comprehensive multimodal corpus designed for both detection and grounding tasks to support the full SFT-RL-Test pipeline. Crucially, we employ Self-Evolving CoT Generation Pipeline to synthesize high-quality CoT annotations for the SFT phase.

(3) **Dynamic Balancing Algorithm ARSPO:** We propose an adaptive reward shaping mechanism that effectively resolves the exploration imbalance in multi-task RL training, significantly boosting performance on high-difficulty tasks.

**Conflict of Interest Disclosure.** The authors declare no conflicts of interest.

## 2. Related Work

**Forgery Detection and Grounding.** With the rapid evolution of generative techniques, detection methodologies have progressed from unimodal paradigms—targeting specific modalities such as images (Borji, 2023; Wang et al., 2023; Sinitsa & Fried, 2024; Lorenz et al., 2023), text (Zhao et al., 2022; Yoo et al., 2023; Mitchell et al., 2023), or videos (Gerstner & Farid, 2022; Agarwal & Farid, 2021)—to bimodal synergies (e.g., image-text (Shen et al., 2025; Zhang et al., 2025a), video-text (Qi et al., 2023)). Nevertheless, the current defensive landscape remains fragmented, notably lacking a unified framework capable of simultaneously processing Image, Video, and Text. This fragmentation stands in stark contrast to the inherent cross-modal entanglement

of contemporary deepfakes, rendering existing models vulnerable to complex, hybrid attacks.

**Reinforcement Learning.** In the realm of Artificial General Intelligence, pioneering works represented by DeepSeek-R1 (DeepSeek-AI, 2025) have demonstrated that RL empowers models with reasoning and generalization capabilities that surpass those achieved by SFT. Optimization techniques such as GRPO (DeepSeek-AI, 2025) and its variants (e.g., GSPO (Zheng et al., 2025), SAPO (Gao et al., 2025)) have been widely adopted to enhance the performance of MLLMs. This capacity for extrapolative generalization aligns perfectly with the core imperatives of Forgery Detection: models must maintain acute reasoning abilities when confronting Out-of-Distribution (OOD) and unseen manipulation techniques. Consequently, we pioneer the integration of RL into the field of forgery detection and grounding.

## 3. Self-Evolving Forensic CoT Generation

High-quality Chain-of-Thought (CoT) data is essential for the RL cold-start phase. However, generating such data for forensics faces an Efficiency-Bias Dilemma: (1) Vanilla generation via closed-source MLLMs yields low-quality reasoning due to their limited domain expertise in forgery detection. (2) While injecting Ground Truth (GT) to force-guide generation introduces Hindsight Bias, where models rationalize answers backward rather than performing genuine deduction, thus impairing RL exploration. To break this impasse, we propose a Self-Evolving CoT Generation strategy consisting of four stages, as shown in Figure 2: Source Data Colletion, Forensic Reasoning Seed Priming, Seed Bootstrapping through Self-Evolution, and Collaborative Hard-CoT Synthesis.

### 3.1. Source Data Collection

We curate a pool of source data by aggregating existing public datasets to cover multiple modalities. For the uni-modal Text domain, we utilize FakeNewsCorpus (Shu et al., 2020) and MCFEND (Li et al., 2024a); for image, we select FakeClue (Wen et al., 2025), LOKI (Ye et al., 2025), and ForgeryNet (He et al., 2021); and for Video, we employ GenVideo (Chen et al., 2024), DVF (Song et al., 2024), and ForgeryNet (He et al., 2021). For the Image-Text modality, we source from SAMM (Shen et al., 2025), MDSM (Zhang et al., 2025a), DGM$^4$ (Shao et al., 2023), and NewsCLIP-pings (Luo et al., 2021). To verify whether the model truly acquires bi-modal detection capabilities from image-text, we intentionally excluded video-text data from this set. Instead, we reserve video-text for out-of-domain zero-shot evaluation, as detailed in Sec. 4.2. Finally, we partition all source data into $D_s$ for SFT dataset generation, $D_r$ for RL data construction, and $D_t$ for in-domain testing.

### 3.2. Forensic Reasoning Seed Priming

We leverage a set of state-of-the-art MLLMs $\mathcal{M} = \{\text{Seed1.6-VL}, \text{Gemini3}, \text{ChatGPT5}\}$ (Guo et al., 2025; Team, 2025; Zhao et al., 2024) to perform inference on a small fraction of samples from $D_s$. Each instance corresponds to a single task (e.g., binary forgery classification). The resultant reasoning path undergoes rigorous ground-truth filtering and consistency verification by a distinct MLLM, aiming to assess the self-consistency and correctness of the reasoning logic (detailed criteria in Appendix H). This process yields a high-fidelity seed dataset $D_s^0$ ($|D_s^0| = 6.7$k). Finally, we perform SFT on Qwen3VL-8B (Yang et al., 2025a) using $D_s^0$, followed by RL on a subset $\bar{D}_r \subset D_r$, resulting in the warm-up policy $\pi_0$.

### 3.3. Seed Bootstrapping through Self-Evolution

To expand data scale and enhance quality further, we design an iterative self-evolution loop. In the $k$-th iteration, we utilize the model from the previous round, $\pi_{k-1}$, to generate CoT reasoning and predictions for $D_s \setminus D_s^0$. Consistent with the previous step, strict filtering is applied based on prediction accuracy against GT. We further employ a SOTA MLLM to validate the CoT generated by $\pi_{k-1}$ (validation protocols align with the Seed priming phase). High-quality samples that pass this verification are merged with the initial seed dataset $D_s^0$ to form the SFT cold-start dataset for the current round, $D_s^k$. To prevent the model from converging to local optima and accumulating distributional biases from the previous iteration, we initialize the SFT process with the base Qwen3VL-8B (rather than $\pi_{k-1}$) on $D_s^k$, followed by RL training on $\bar{D}_r$, resulting in $\pi_k$. This process terminates after three iterations, yielding $D_s^3$.

### 3.4. Collaborative Hard-CoT Synthesis

Despite the efficacy of the self-evolution loop, the model continues to struggle with finding correct reasoning paths for certain long-tail hard samples (i.e., samples that remain incorrectly predicted even after multiple iterations). To complete the data distribution and avoid introducing difficulty bias, we design a Multi-Agent Collaborative Generation workflow. First, a randomly selected MLLM $\in \mathcal{M}$ generates a CoT conditioned on the known GT. Second, a distinct MLLM $\in \mathcal{M}$ acts as a "Refiner," constrained to rewrite the CoT to simulate a natural deduction process as if the answer were unknown, while maintaining logical correctness (See Appendix H). Finally, a third MLLM $\in \mathcal{M}$ evaluates the refined CoT using the same criteria as in Sec. 3.2, yielding dataset $D_s^h$.

### 3.5. Dataset Statistics

Through the aforementioned process, we obtain the **Full-Spectrum Forensic Reasoning (FSFR)** dataset, a comprehensive multimodal corpus that integrates primary social media modalities for both detection and grounding tasks. FSFR consists of two parts: $\text{FSFR}_{\text{sft}} = D_s^h \cup D_s^3$ for cold-start SFT with high-quality CoT annotations and $\text{FSFR}_{\text{rl}} = D_r \setminus \bar{D}_r$ for reinforcement learning, and $D_t$ for performance evaluation. We exclude the warm-up subset $\bar{D}_r$ from the RL stage. $\text{FSFR}_{\text{sft}}$ contains 73k samples, where hard samples ($D_s^h$) account for approximately 15%; $\text{FSFR}_{\text{rl}}$ comprises 110k samples. FSFR covers Text, Image, Video, and Image-Text modalities, supporting four core tasks: binary forgery classification, and tampering localization across image (spatial), text (semantic), and video (temporal) domains. $D_t$ denotes the union of the test sets from aforementioned datasets in Sec. 3.1, comprising approximately 700k samples for reliable performance evaluation.

## 4. ARSPO: Balanced Multi-Task RL

While FSFR provides the necessary foundations for cold-start of RL, we observe that conventional RL algorithms suffer from difficulty imbalance, where optimization is dominated by simpler sub-tasks as shown in Table 1b. To address this, we propose ARSPO (Adaptive Reward Shaping Policy Optimization). ARSPO dynamically balances the learning process by adaptively scaling reward signals across tasks of varying difficulty, ensuring consistent policy improvement across the entire forensic spectrum.

### 4.1. Theoretical Analysis and Motivation

To elucidate how different tasks differentially impact the model's update trajectory, we unify the optimization objectives of GRPO and its variants (e.g., GSPO, SAPO) (Gao et al., 2025). Let $\theta$ denote the model parameters, $\mathcal{D}$ the

total dataset, and $\mathcal{D}_k$ the subset corresponding to the $k$-th task. For each query $q \in \mathcal{D}$, the model samples $G$ responses $\{y_i\}_{i=1}^G$. Let $\pi_\theta$ and $\pi_{\theta_{\text{old}}}$ represent the current and old policies, respectively. The RL algorithm maximizes the following objective:

$$\mathcal{J}(\theta) = \sum_{k=1}^K \frac{|\mathcal{D}_k|}{|\mathcal{D}|} \mathbb{E}_{\substack{q \sim \mathcal{D}_k, \\ \{y_i\} \sim \pi_{\theta_{\text{old}}}}} \left[ \frac{1}{G} \sum_{i=1}^G \frac{1}{|y_i|} \sum_{t=1}^{|y_i|} f_{i,t}(r_{i,t}(\theta)) \hat{A}_{i,k} \right] \quad (1)$$

where $r_{i,t} = \frac{\pi_\theta(y_{i,t}|q,y_{i,<t})}{\pi_{\theta_{\text{old}}}(y_{i,t}|q,y_{i,<t})}$ is the probability ratio, and $\hat{A}_{i,k} = \frac{A_{i,k}-\mu}{\sigma}$ is the normalized advantage, with $\mu$ and $\sigma$ being the mean and standard deviation of the rewards $\{A_{i,k}\}_{i=1}^G$. Different RL algorithms correspond to different forms of $f_{i,t}(\cdot)$ (see Appendix B for details).

Differentiating Eq. (1) yields the log-policy gradient:

$$\nabla_\theta \mathcal{J}(\theta) = \sum_{k=1}^K \frac{|\mathcal{D}_k|}{|\mathcal{D}|} \mathbb{E} \left[ \frac{1}{G} \sum_{i=1}^G \frac{1}{|y_i|} \sum_{t=1}^{|y_i|} f'_{i,t}(r_{i,t}) r_{i,t} \nabla_\theta \log \pi_\theta(y_{i,t}|q, y_{i,<t}) \hat{A}_{i,k} \right] \quad (2)$$

To probe the deeper mechanisms of training dynamics—specifically how task difficulty dictates gradient direction—we take the derivative of Eq. 2 with respect to $\theta$. $\nabla_\theta \mathcal{J}(\theta)$ examines how parameter shifts induce changes in gradient magnitude, thereby revealing convergence trends.

Let $W_{i,t}(\theta) = f'_{i,t}(r_{i,t}) r_{i,t} \nabla_\theta \log \pi_\theta(y_{i,t}|q, y_{i,<t})$ be the gradient term related to the geometric properties of the policy itself. Here, we explicitly model the mapping between model capability and task metrics. Let $H_k(\theta, q, \tau)$ denote the task-specific performance metric (e.g., IoU in visual localization) achievable by the model's response to input $q$ under parameters $\theta$, such that $x_{i,k} = H_k(\theta, q, \tau)$ (where $\tau$ represents diversity controlled by hyperparameters like temperature). We define $g_k(\cdot)$ as the function mapping this metric to the raw reward value, i.e., $A_{i,k} = g_k(x_{i,k})$. Differentiating Eq. (2), we obtain the change rate of the gradient:

$$\frac{d}{d\theta}(\nabla_\theta \mathcal{J}(\theta)) = \sum_{k=1}^K \frac{|\mathcal{D}_k|}{|\mathcal{D}|} \mathbb{E} \left[ \frac{1}{G} \sum_{i=1}^G \frac{1}{|y_i|} \sum_{t=1}^{|y_i|} \frac{d}{d\theta}\left(W_{i,t}(\theta) \hat{A}_{i,k}\right) \right] \quad (3)$$

The expansion of the core term is:

$$\frac{d}{d\theta}\left(W_{i,t}(\theta) \hat{A}_{i,k}\right) = W'_{i,t}(\theta) \hat{A}_{i,k} + W_{i,t}(\theta) \\ \times \frac{g'_k(H_k(\theta,q,\tau))}{G\sigma} \left[(G-1) - \hat{A}_{i,k}^2\right] H'_k(\theta,q,\tau) \quad (4)$$

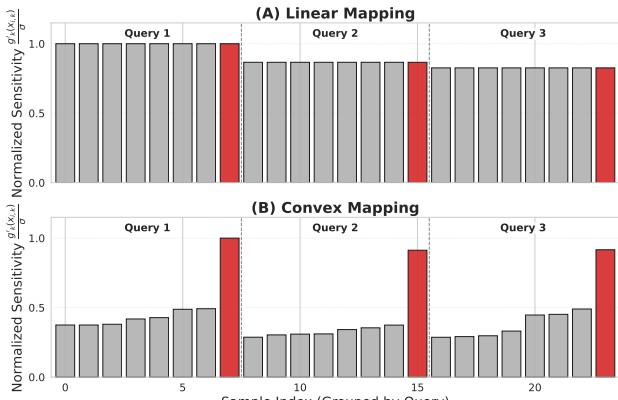

*Figure 3.* Impact of different reward mapping functions $g_k(\cdot)$ on the gradient sensitivity term $\frac{g'_k(x_{i,k})}{\sigma}$ in Eq. 4. (A) Linear mapping with $g_k(x) = ax$; (B) Convex mapping with $g_k(x) = e^{ax}$ (where $a = 3$). Red bars indicate superior responses with higher rewards $A_{i,k}$ within each query group. The comparison demonstrates that **convex mapping significantly amplifies the gradient contribution of superior responses**, whereas linear mapping results in nearly uniform sensitivity across responses of varying quality.

Eq.4 profoundly reveals the source of the "Matthew Effect" in multi-task RL. The second term on the right-hand side comprises two sensitivity components: task difficulty sensitivity $H'_k(\theta, q, \tau)$ and reward function sensitivity $g'_k(\cdot)$. $H'_k(\theta, q, \tau)$ characterizes the extent to which minute parameter changes improve task metrics. For high-difficulty tasks, the model often resides on a performance plateau where metric improvements are negligible, resulting in $H'_k(\theta, q, \tau)$ is small. Consequently, even if normalization balances reward scales, simple tasks (with larger $H'_k$) dominate the second-order rate of change (i.e., the "acceleration") of the gradient, effectively hijacking the primary direction of model updates.

However, Eq.4 also points to a solution: modulating the gradient via the derivative term $g'_k(H_k(\theta, q, \tau))$ of a non-linear reward mapping function $g_k(\cdot)$. ***If we design $g_k(\cdot)$ with a progressively steepening slope as the performance metric improves, the gradient contribution for superior responses (larger $A_{i,k}$ compared to other rewards in the same group) is significantly amplified.*** This suggests that a well-designed reward function can effectively compensate for gradient attenuation caused by task difficulty, incentivizing the model to break through performance bottlenecks in complex tasks. Figure 3 illustrates the behavior of two different $g_k(\cdot)$ functions given an identical set of responses.

Notably, this analysis naturally extends to scenarios involving Reward Models by formulating $A_{i,k} = g_k(\pi_{\theta_{rm}}(x_{i,k}))$. In such cases, $\pi_{\theta_{rm}}$ denotes the Reward Model, leaving the final derivation unaltered. Please refer to the Appendix C for the full derivation of Eqs. 1-4 and differentiability discussions in D, E. For a detailed analysis of Eq. 4, please refer to Appendix F.

**Algorithm 1** Dynamic Coefficient Adjustment

---

**Initialize:** Coefficients $l_{k,s} \leftarrow 1$ for all tasks $k$; Baselines $B_k \leftarrow$ moving average.

**for** global step $s = 1, \ldots, S_{max}$ **do**

    Sample batch data $\mathcal{B} = \{(q, y)\}$ from dataset $\mathcal{D}$.

    **if** $s > T_{warm}$ **and** $s \bmod T == 0$ **then**

        Compute current mean $\mu_k$ over $[s - T, s]$ and past mean $\mu_{past}$ over $[s - 3T, s - T]$.

        Calculate gain $\Delta_{total,k} = (\mu_k - B_k)/B_k$.

        Identify Laggard: $k_{lag} \leftarrow \arg\min_k \Delta_{total,k}$.

        **for** each task $k \in$ Tasks **do**

            $\delta_{recent} \leftarrow \mu_k - \mu_{past}$

            **if** $\delta_{recent} > \epsilon_{mom}$ **then**

                **continue**

            **else if** $\delta_{recent} < -\epsilon_{rescue}$ **then**

                $l_{k,s} \leftarrow l_{k,s} \cdot \alpha_{boost}$

            **else if** $\Delta_{total,k} > \tau_{high}$ **then**

                $l_{k,s} \leftarrow \max(l_{k,s} \cdot \alpha_{decay}, 1)$

            **else if** $k == k_{lag}$ **then**

                $l_{k,s} \leftarrow \min(l_{k,s} \cdot \alpha_{boost}, 4)$

            **end if**

        **end for**

    **end if**

    **Rescaling:** $l_{k,s} \leftarrow l_{k,s} / \min(\{l_{j,s}\}_{j=1}^K)$

    **Policy Update:** Update $\theta$ using Eq. (6) with $\{l_{k,s}\}$.

**end for**

---

## 4.2. ARSPO: Adaptive Reward Shaping Policy Optimization

ARSPO introduces two key innovations: 1. Guided by the analysis in Sec. 4.1, we design distinct $g_k(\cdot)$ functions for different tasks; 2. To further balance the disparity in task difficulty adaptively during training, we propose a dynamic coefficient adjustment algorithm, introducing a dynamic coefficient $l_{k,s}$ (representing the coefficient for task $k$ at training step $s$). The optimization objective of ARSPO is:

$$
\mathcal{J}_{arspo}(\theta) = \sum_{k=1}^{K} \frac{|\mathcal{D}_k|}{|\mathcal{D}|} \mathbb{E}_{\substack{q \sim \mathcal{D}_k, \\ \{y_i\} \sim \pi_{\theta_{\text{old}}}}} \left[ \frac{l_{k,s}}{G} \sum_{i=1}^{G} \frac{1}{|y_i|} \sum_{t=1}^{|y_i|} f_{i,t}(r_{i,t}(\theta)) \hat{A}_{i,k} \right] \quad (5)
$$

Differentiating Eq. 5 yields the log-policy gradient:

$$
\nabla_\theta \mathcal{J}_{arspo}(\theta) = \sum_{k=1}^{K} \frac{|\mathcal{D}_k|}{|\mathcal{D}|} \mathbb{E} \left[ \frac{l_{k,s}}{G} \sum_{i=1}^{G} \frac{1}{|y_i|} \sum_{t=1}^{|y_i|} f'_{i,t}(r_{i,t}) r_{i,t} \nabla_\theta \log \pi_\theta(y_{i,t}|q, y_{i,<t}) \hat{A}_{i,k} \right] \quad (6)
$$

**Task-Based Reward Mapping Function.** For binary classification, we select the identity mapping $g_k(x) = x$. For the

three fine-grained tasks: visual tampering localization, text tampering localization, and video temporal tampering localization, we select $g_k(x) = e^{a_k x}$, which possesses a steeper slope to amplify gradients in high-performance regions.

**Dynamic Coefficient Adjustment.** As outlined in Algorithm 1, this mechanism periodically monitors learning trends and adjusts $l_{k,s}$ accordingly:

*- Initialization and Baseline Collection.* During the warm-up phase (global step $s < T_{warm}$), we initialize all task coefficients $l_{k,s}$ to 1 and record the average metrics to establish a baseline $B_k$. For instance, $B_k$ corresponds to average Accuracy for classification and average IoU for visual localization. This establishes a reference point for measuring subsequent progress. Crucially, $B_k$ is frozen once the warm-up concludes, remaining constant for all steps $s \geq T_{warm}$.

*- Status Estimation and Laggard Identification.* Post-warm-up, coefficients are adjusted every $T$ steps. At step $s$, we compute the current mean $\mu_k$ over $[s - T, s]$. We define the total relative gain as $\Delta_{total,k} = (\mu_k - B_k)/B_k$. To prevent the "bucket effect," we identify the "Laggard" task with the slowest progress: $k_{lag} = \arg\min_k \Delta_{total,k}$. We aslo record the historical mean $\mu_{past}$ over $[s - 3T, s - T]$ for further process.

*- Priority-Based Dynamic Adjustment Strategy.* We employ a hierarchical logic to update $l_{k,s}$ based on the recent trend $\delta_{recent} = \mu_k - \mu_{past}$. The symbols $\epsilon_{mom}$, $\epsilon_{rescue}$, $\tau_{high}$, $\alpha_{decay}$, and $\alpha_{boost}$ denote hyper-parameters, whose specific values are detailed in Appendix Table 10.

▷ *Momentum Protection:* If $\delta_{recent} > \epsilon_{mom}$, the task is in a rapid ascent phase. We maintain the current $l_{k,s}$.

▷ *Regression Rescue:* If $\delta_{recent} < -\epsilon_{rescue}$, indicating significant performance degradation, we multiply $l_{k,s}$ by a boost factor $\alpha_{boost}$ for emergency weighting.

▷ *High-Performance Decay:* If $\Delta_{total,k} > \tau_{high}$, implying the task is mastered, we multiply $l_{k,s}$ by a decay factor $\alpha_{decay}$ (clamped to $\geq 1$) to reduce its gradient dominance while retaining base importance.

▷ *Laggard Support:* If none of the above trigger and the task is identified as $k_{lag}$, we multiply its coefficient by $\alpha_{boost}$ to accelerate learning on this difficult task.

*- Parameter Update.* At each step $s$, we rescale the latest coefficients $\{l_{k,s}\}$ by the minimum coefficient across tasks, then use them in Eq. 6 to compute the weighted gradient and update $\theta$, enabling a smooth and dynamically balanced progression from simple to complex tasks.

## 5. Experiments

Please refer to the Appendix A for Implementation Details.

*Table 2.* Performance Comparison on In-Domain test dataset $D_t$, our method achieves outstanding performance across diverse modalities.

| Method | Binary Classification(ACC) | | | | Image Loc. (IoU) | Text Loc. (F1) | Video Loc. (tIoU) |
| --- | --- | --- | --- | --- | --- | --- | --- |
| | Text | Image | Video | Text-Image | | | |
| *Multi-modal Large Language Models* | | | | | | | |
| Qwen3VL-235B (Yang et al., 2025a) | 89.23 | 65.32 | 70.80 | 58.42 | 16.37 | 31.25 | 21.43 |
| Llama3.2-90B (Team, 2024) | 83.47 | 60.34 | 63.28 | 55.91 | 14.85 | 25.99 | 19.54 |
| *Text-Image Detection Methods* | | | | | | | |
| HAMMER (Shao et al., 2023) | — | — | — | 71.23 | 48.53 | 40.86 | — |
| FKA-Owl (Liu et al., 2024) | — | — | — | 72.08 | — | — | — |
| AMD (Zhang et al., 2025a) | — | — | — | 65.34 | 35.91 | 37.64 | — |
| *Text Detection Methods* | | | | | | | |
| Bert (Devlin et al., 2019) | 61.45 | — | — | — | — | 35.91 | — |
| LUKE (Yamada et al., 2020) | 63.72 | — | — | — | — | 38.28 | — |
| *Image Detection Methods* | | | | | | | |
| $D^3$ (Yang et al., 2025b) | — | 88.46 | — | — | — | — | — |
| Fake-VLM (Wen et al., 2025) | — | 90.39 | — | — | — | — | — |
| *Video Detection Methods* | | | | | | | |
| FakeSV-VLM (Wang et al., 2025) | — | — | **98.81** | — | — | — | — |
| Video-R1 (Feng et al., 2025) | — | — | 96.44 | — | — | — | — |
| **OmniVL-Guard (Ours)** | **96.20** | **93.12** | 98.58 | **75.52** | **54.26** | **63.78** | **59.22** |
| Δ **(vs Best)** | +6.97 | +2.73 | -0.23 | +3.44 | +5.73 | +22.92 | +37.79 |

*Table 3.* Performance Comparison on Out-Of-Domain Benchmarks (Binary Classification).

| Modality | Dataset | MLLMs | | Image-Text Methods | | | Text Methods | | Image Methods | | Video-Text Methods | | Ours |
| --- | --- | --- | --- | --- | --- | --- | --- | --- | --- | --- | --- | --- | --- |
| | | Qwen3VL | Llama3.2 | HAMMER | FKA-Owl | AMD | Bert | LUKE | $D^3$ | Fake-VLM | FakeSV-VLM | Video-R1 | Ours |
| **Text** | ISOT | 88.74 | 81.91 | — | — | — | 33.95 | 35.78 | — | — | — | — | **93.69** |
| **Image** | CASIA2.0 | 60.88 | 58.17 | — | — | — | — | — | 49.20 | 51.22 | — | — | **63.64** |
| **Text-Image** | MMFakeBench | 57.40 | 54.14 | 61.83 | 62.32 | 59.11 | — | — | — | — | — | — | **79.38** |
| **Text-Video** | FakeSV | 61.22 | 56.93 | — | — | — | — | — | — | — | 52.59 | 55.32 | **63.55** |

**Benchmarks and Baselines.** We categorize benchmarks into In-Domain (test sets from Sec. 3.1) and Out-Of-Domain (OOD) testing, designed to evaluate zero-shot generalization. The OOD suite includes ISOT (Ahmed et al., 2017) (text), CASIA2.0 (Yadav & Vishwakarma, 2024) (image), MM-FakeBench (Liu et al., 2025) (image-text), and FakeSV (Qi et al., 2023) (video-text); see Appendix L for details. For baselines, general MLLMs (e.g., Qwen3VL-235B (Yang et al., 2025a)) are evaluated in a zero-shot setting. In contrast, domain-specific methods and our model are trained on FSFR and then directly applied to OOD datasets without further fine-tuning to assess cross-domain robustness.

**5.1. Performance Comparison**

**In-Domain Comparison.** As presented in Table 2, OmniVL-Guard achieves leading performance across all multimodal detection and localization tasks. Compared to general MLLMs and single-modality expert methods (e.g., Fake-VLM, $D^3$), our model significantly overcomes perfor-

mance bottlenecks in challenging fine-grained localization tasks while maintaining highly competitive binary classification accuracy. Notably, in video temporal localization (tIoU) and text localization (F1), OmniVL-Guard yields substantial improvements of 37.79% and 22.92%, respectively.

**Out-Of-Domain Zero-Shot.** As shown in Table 3, OmniVL-Guard achieves strong OOD zero-shot performance, outperforming domain-specific models such as FKA-Owl that tend to overfit. By using RL-enhanced reasoning to capture intrinsic forgery cues, our method reaches accuracies of 79.38% and 63.55% on Text-Image and Text-Video tasks, respectively. These results show that self-evolving CoT and ARSPO effectively improve generalization to unseen forgeries.

**5.2. Ablation Study**

**Ablation of ARSPO Components.** As shown in Table 4, our ablation study systematically validates the necessity of

*Table 4.* Ablation study of different components using **Image Loc.** (IoU), **Text Loc.** (F1), and **Video Loc.** (tIoU) metrics. The rightmost column shows the average performance improvement (Δ AVG) across all tasks compared to the SFT baseline.

| SFT | SAPO | TBRMF | DCA | \| | Text | Image | Video | Text-Image | \| | Image | Text | Video | \| Δ AVG |
|-----|------|-------|-----|----|------|-------|-------|------------|----|-------|------|-------|---------|
| | | | | | | **Binary Classification** | | | | | **Localization** | | |
| ✓ | | | | \| | 83.73 | 32.41 | 33.82 | 63.57 | \| | 51.08 | 44.67 | 33.08 | \| – |
| ✓ | ✓ | | | \| | 95.71 | 93.25 | **99.10** | 74.92 | \| | 51.24 | 54.33 | 44.10 | \| +24.33 |
| ✓ | ✓ | ✓ | | \| | 95.93 | 92.99 | 98.57 | **75.85** | \| | 53.21 | 61.37 | 49.38 | \| +26.42 |
| ✓ | ✓ | | ✓ | \| | **96.99** | **93.57** | 99.04 | 74.97 | \| | 52.95 | 59.88 | 53.49 | \| +26.93 |
| ✓ | ✓ | ✓ | ✓ | \| | 96.20 | 93.12 | 98.58 | 75.52 | \| | **54.26** | **63.78** | **59.22** | \| **+28.33** |

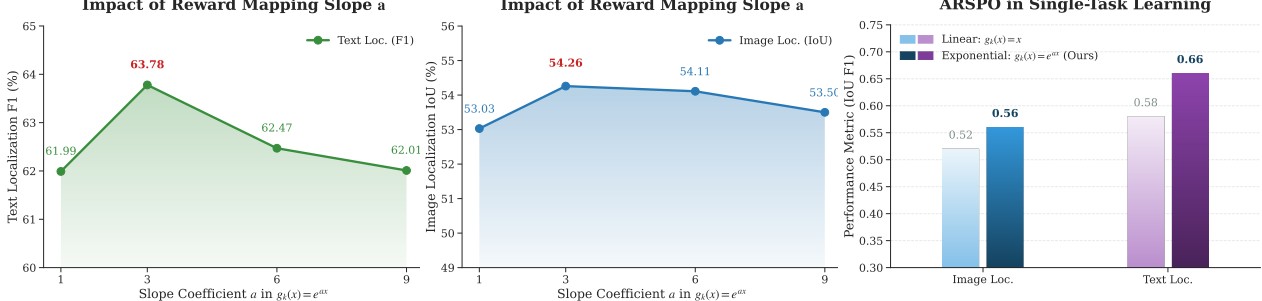

*(a)* Text Tampering Localization  *(b)* Image Tampering Localization  *(c)* Single Task Scenarios

*Figure 4.* **Ablation Study on Reward Mapping Steepness and Single-task Scenarios.** (a) Impact of parameter $a$ on text localization; (b) Impact of parameter $a$ on image localization; (c) Performance comparison of reward mapping functions in single-task settings.

each core component within the ARSPO framework. The cold-start model, trained solely via SFT on FSFR$_\text{sft}$, exhibits suboptimal performance across all tasks. Upon introducing the SAPO, the model demonstrates emergent reasoning capabilities, achieving a substantial leap in binary classification accuracy and establishing the role of RL in general forgery detection; however, optimization bottlenecks persist in challenging localization tasks. After incorporating the Task-Based Reward Mapping Function (TBRMF) and Dynamic Coefficient Adjustment (DCA), the gradient signals for difficult tasks are effectively amplified, and training weights are dynamically balanced. Ultimately, the complete ARSPO framework, significantly elevates localization performance (e.g., Video Loc. rises to $59.22\%$).

**Impact of Reward Mapping Steepness.** Figures 4a-4b indicate that performance peaks at $a = 3$, confirming the utility of non-linear rewards. Beyond this point, performance declines because overly steep mappings amplify aleatoric noise. This induces "reward overfitting," causing the policy to trap in local optima by learning noise features. Consequently, $a = 3$ provides the necessary balance between signal amplification and robust training.

### 5.3. Study of ARSPO in Single-Task Scenarios

As shown in Figure 4c, to validate the effectiveness of AR-SPO in single-task scenarios, we compare the effects of linear versus exponential reward functions. Experimental

results compellingly demonstrate that even in the absence of inter-task resource competition, the exponential mapping strategy achieves significant performance leaps in image and text localization tasks, reaching scores of $0.56$ $(+4\%)$ and $0.66$ $(+8\%)$, respectively. This finding reveals that the advantage of ARSPO lies not merely in "balancing," but fundamentally in the "reshaping" of gradient signals via non-linear transformation.

### 5.4. Additional Analysis of Self-Evolving CoT

**Self-evolution saturation.** To determine the optimal termination point for Self-Evolving CoT Generation, we investigate the performance trajectory across iterations $k$. We hypothesize that the self-evolution process eventually reaches a saturation point where the teacher policy $\pi_k$ no longer provides stronger reasoning signals than $\pi_{k-1}$, and thus cannot yield further quality gains in the generated dataset $D_s^k$. Specifically, we construct a fourth-round dataset $D_s^4$ using $\pi_3$, combine it with the hard-sample set $D_s^h$, and train the full OmniVL-Guard framework via the same SFT+ARSPO recipe. We then compare this setting with our standard configuration, $D_s^3 \cup D_s^h$.

As shown in Table 5, the performance differences between the third and fourth iterations are statistically negligible. For instance, Image Localization shifts only from $54.26$ to $54.41$, while Video Localization slightly fluctuates by $-0.15$. This evidence suggests that the quality of the CoT

*Table 5.* Performance comparison between stopping at Iteration 3 ($D_s^3$) and extending to Iteration 4 ($D_s^4$). The results indicate that performance plateaus at $k = 3$, justifying the termination of the self-evolution loop to conserve computational resources.

| Training Set | Binary Classification (ACC) | | | | Localization | | |
|---|---|---|---|---|---|---|---|
| | Text | Image | Video | Text-Image | Image (IoU) | Text (F1) | Video (tIoU) |
| $D_s^3 \cup D_s^h$ | 96.20 | 93.12 | 98.58 | 75.52 | 54.26 | 63.78 | 59.22 |
| $D_s^4 \cup D_s^h$ | **96.35** | 93.01 | 98.48 | **75.60** | **54.41** | **63.87** | 59.07 |
| $\Delta$ | +0.15 | -0.11 | -0.10 | +0.08 | +0.15 | +0.09 | -0.15 |

*Table 6.* Ablation study on the necessity of Collaborative Hard-CoT Synthesis. We report the performance comparison between using the full dataset and removing the synthesized hard samples ($D_s^h$).

| Method | Binary Classification (ACC) | | | | Localization | | |
|---|---|---|---|---|---|---|---|
| | Text | Image | Video | Text-Image | Image (IoU) | Text (F1) | Video (tIoU) |
| **Full FSFR$_{\text{sft}}$** | **96.20** | **93.12** | **98.58** | **75.52** | **54.26** | **63.78** | **59.22** |
| w/o Hard-CoT | 92.16 | 86.90 | 91.34 | 70.33 | 50.44 | 56.31 | 54.14 |
| $\Delta$ | -4.04 | -6.22 | -7.24 | -5.19 | -3.82 | -7.47 | -5.08 |

data generated by $\pi_3$ has reached a bottleneck, and further iterations do not yield distinguishable gains in downstream forensic tasks. Consequently, we terminate the self-evolution process at $k = 3$ to strike an optimal balance between model performance and training efficiency.

**Effect of Collaborative Hard-CoT Synthesis.** To validate the necessity of Collaborative Hard-CoT Synthesis, we ablate the hard-sample subset $D_s^h$ from the SFT cold-start data. We compare the full setting, where the model is fine-tuned on FSFR$_{\text{sft}} = D_s^3 \cup D_s^h$ and then trained with ARSPO, against a w/o Hard-CoT setting, where the model is fine-tuned only on the self-evolved dataset $D_s^3$ before the same ARSPO stage.

Table 6 shows that removing hard samples leads to consistent performance degradation. The impact is particularly significant for fine-grained localization: Text Localization drops by $7.47\%$ and Video Temporal Localization drops by $5.08\%$. This indicates that the self-evolution loop alone tends to saturate on easier samples, while the remaining failure cases often contain subtle semantic conflicts or complex temporal inconsistencies that require the high-quality reasoning logic provided by Collaborative Hard-CoT Synthesis. Binary classification also decreases notably, such as a $5.19\%$ drop on Image-Text detection, showing that $D_s^h$ is useful not only for grounding but also for robust forgery detection in complex multimodal scenarios. Overall, Collaborative Hard-CoT Synthesis complements self-evolution by covering the long-tail distribution of difficult forgery types and preventing the model from overfitting to simple patterns.

**Hindsight bias in GT-guided CoT.** We also examine a simpler alternative that directly injects ground-truth labels when prompting strong MLLMs to generate CoT annotations. Although this strategy can guarantee label correct-

ness, it yields reasoning paths that often rationalize the answer backward rather than discover forensic evidence step by step. Under the same SFT+ARSPO recipe, Direct GT Injection obtains 92.15, 84.47, 91.30, and 69.12 on Text, Image, Video, and Text-Image classification, and 51.50, 52.84, and 46.10 on Image, Text, and Video localization. Self-Evolving CoT improves these results to 96.20, 93.12, 98.58, 75.52, 54.26, 63.78, and 59.22, respectively. The resulting gaps are substantial across all tasks ($-4.05$, $-8.65$, $-7.28$, $-6.40$, $-2.76$, $-10.94$, and $-13.12$), especially for reasoning-intensive localization. These results suggest that CoT quality cannot be reduced to final-answer correctness. For forensic grounding, the intermediate reasoning must preserve an exploratory evidence-gathering process so that subsequent RL can learn robust localization behavior instead of superficial answer rationalization.

### 5.5. More Discussion

We further discuss the generalizability and potential bias of ARSPO and FSFR in Appendix J.

## 6. Conclusion

This work presents OmniVL-Guard, a unified framework designed to tackle the complexities of interleaved multimodal forgery detection and grounding. By identifying the difficulty bias that hinders joint optimization, we develop a Self-Evolving CoT Generation and the Adaptive Reward Scaling Policy Optimization (ARSPO) strategy to ensure balanced learning across classification and localization tasks. Extensive evaluations on diverse benchmarks demonstrate that our approach significantly surpasses existing state-of-the-art methods and maintains robust zero-shot generalization in diverse out-of-domain scenarios. These findings underscore the importance of addressing task imbalances in multi-modal forensics and provide a scalable foundation for future research in content integrity.

## Impact Statement

This work presents OmniVL-Guard to improve the integrity of the digital information ecosystem by detecting complex multimodal misinformation. While its reasoning capabilities offer substantial societal benefits, we recognize potential dual-use risks. To mitigate them, the dataset will be released only for academic use through a rigorous application-and-review process.

## Acknowledgements

This work was funded by the National Natural Science Foundation of China (No. 62572166, 62302140, 62502144, 62502142, 62573399) and the Natural Science Foundation

of Anhui Province (No. 2508085QF226). The computation is completed on the HPC Platform of Hefei University of Technology.

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

# A. Implementation Details

We train our model on 16 NVIDIA H100 (96GB) GPUs, utilizing Qwen3VL-8B as the backbone. We employ LoRA (Hu et al., 2022) for both the SFT and RL stages. During the SFT phase, the global batch size is set to 120 with a learning rate of $2.5 \times 10^{-4}$. In the RL phase, the global batch size is set to 27 with a learning rate of $1.5 \times 10^{-5}$. For RL training, we implement ARSPO based on SAPO, setting the group size to 8 and fixing the KL regularization coefficient $\beta_{\text{KL}}$ at 0.01. For the three fine-grained tasks—image tampering localization, text tampering localization, and video temporal tampering localization—the parameter $a$ in the reward mapping function is consistently set to 3.

## A.1. Reward Function Configuration

In the RL phase, the total reward $R$ for a generated response $y$ given input $x$ and ground truth $g$ is composed of three distinct components: the task-specific performance score ($R_{\text{task}}$), the structural format reward ($R_{\text{fmt}}$), and a repetition penalty ($R_{\text{rep}}$). The final reward is calculated as:

$$R(y, g) = R_{\text{task}}(y, g) + R_{\text{fmt}}(y) + R_{\text{rep}}(y) \tag{7}$$

### 1. Task-Specific Performance Reward ($R_{\text{task}}$)

We employ distinct metrics for different forensic tasks. To encourage the model to strive for high-precision grounding rather than mediocre overlap, we apply an exponential mapping function to continuous metrics (IoU, F1).

- **Binary Forgery Classification:** For the binary classification task, we utilize the raw model confidence directly to provide a continuous signal. Let $x$ be the *logit* value corresponding to the ground truth label ("Real" or "Fake"). The reward is defined as an identity mapping:

$$R_{\text{task}}^{\text{cls}} = x \tag{8}$$

  Unlike the discrete indicator function, this formulation incentivizes the model to maximize the confidence margin of the correct prediction.

- **Tampering Localization (Image, Text, Video):** For fine-grained localization tasks, we compute a raw metric $m \in [0, 1]$ specific to the modality: Intersection over Union (IoU) for image bounding boxes, F1-score for text indices, and Temporal IoU (tIoU) for video segments. The raw metric is mapped to a normalized reward using an exponential function with a scaling factor $\alpha = 3$:

$$R_{\text{task}}^{\text{loc}}(m) = \frac{e^{\alpha \cdot m} - 1}{e^{\alpha} - 1} \tag{9}$$

  This convex mapping ($\alpha = 3$) amplifies the reward signal for high-quality responses (e.g., $m > 0.7$) while suppressing rewards for low-quality overlaps.

### 2. Format Compliance Reward ($R_{\text{fmt}}$)

To enforce the Chain-of-Thought (CoT) structure, we employ a strict regular expression check. The model receives a positive reward if and only if the output strictly follows the pattern `<think>...</think><answer>...</answer>` without trailing characters:

$$R_{\text{fmt}} = \begin{cases} 0.2 & \text{if format is valid} \\ 0.0 & \text{otherwise} \end{cases} \tag{10}$$

### 3. Repetition Penalty ($R_{\text{rep}}$)

To prevent degenerate generation loops, we implement an N-gram repetition penalty. We calculate the ratio of unique N-grams to total N-grams in the generated text. With a set hyperparameters of $N = 3$ and a maximum penalty coefficient $\lambda_{\text{pen}} = -1.0$, the penalty is defined as:

$$R_{\text{rep}} = \lambda_{\text{pen}} \times \left( 1 - \frac{|S_{\text{unique}}^N|}{N_{\text{total}}} \right) \tag{11}$$

where $|S_{\text{unique}}^N|$ is the count of unique 3-grams and $N_{\text{total}}$ is the total number of 3-grams. This term introduces a negative reward proportional to the redundancy of the text.

## B. Specific Formulations of RL Algorithms

The unified mathematical formulation for GRPO and its variants, GSPO and SAPO, follows the objective function presented in Eq. 1 of Sec. 4:

$$\mathcal{J}(\theta) = \sum_{k=1}^{K} \frac{|\mathcal{D}_k|}{|\mathcal{D}|} \mathbb{E}_{q \sim \mathcal{D}_k, \{y_i\} \sim \pi_{\theta_{\text{old}}}} \left[ \frac{1}{G} \sum_{i=1}^{G} \frac{1}{|y_i|} \sum_{t=1}^{|y_i|} f_{i,t}(r_{i,t}(\theta)) \hat{A}_{i,k} \right] \tag{12}$$

For **GRPO**, the weighting function $f_{i,t}^{\text{grpo}}(\cdot)$ is defined as:

$$f_{i,t}^{\text{grpo}}(r_{i,t}(\theta); \widehat{A}_{i,k}) = \begin{cases} \min(r_{i,t}(\theta), 1+\varepsilon), & \widehat{A}_{i,k} > 0, \\ \max(r_{i,t}(\theta), 1-\varepsilon), & \widehat{A}_{i,k} \leq 0, \end{cases} \tag{13}$$

For **GSPO**, the function $f_{i,t}^{\text{gspo}}(\cdot)$ is formulated as follows:

$$f_{i,t}^{\text{gspo}}(r_{i,t}(\theta); \widehat{A}_{i,k}) \equiv f_{i,t}^{\text{seq}}(s_{i,t}(\theta); \widehat{A}_{i,k}) = \begin{cases} \min(s_{i,t}(\theta), 1+\varepsilon), & \widehat{A}_{i,k} > 0, \\ \max(s_{i,t}(\theta), 1-\varepsilon), & \widehat{A}_{i,k} \leq 0. \end{cases} \tag{14}$$

where the sequence-level importance ratio $s_{i,t}(\theta)$ is derived as:

$$s_i(\theta) = \left( \frac{\pi_\theta(y_i \mid q)}{\pi_{\theta_{\text{old}}}(y_i \mid q)} \right)^{\frac{1}{|y_i|}} = \exp \left( \frac{1}{|y_i|} \sum_{t=1}^{|y_i|} \log r_{i,t}(\theta) \right), s_{i,t}(\theta) = \text{sg}[s_i(\theta)] \cdot \frac{\pi_\theta(y_{i,t} \mid q, y_{i,<t})}{\text{sg}[\pi_\theta(y_{i,t} \mid q, y_{i,<t})]} \tag{15}$$

Here, $\text{sg}(\cdot)$ denotes the stop-gradient operator.

For **SAPO**, the function $f_{i,t}^{\text{sapo}}(\cdot)$ adopts the following form:

$$f_{i,t}^{\text{sapo}}(r_{i,t}(\theta)) = \frac{4}{\tau_i} \sigma(\tau_i(r_{i,t}(\theta) - 1)), \quad \tau_i = \begin{cases} \tau_{\text{pos}}, & \widehat{A}_{i,k} > 0, \\ \tau_{\text{neg}}, & \widehat{A}_{i,k} \leq 0, \end{cases} \tag{16}$$

where $\tau_{\text{neg}}$ and $\tau_{\text{pos}}$ denote the temperature parameters for positive and negative tokens, respectively, and $\sigma(x) = \frac{1}{1+e^{-x}}$ represents the sigmoid function.

## C. Detailed Derivations for Sec. 4

### C.1. Derivation from Eq. 1 to Eq. 2

$$\nabla_\theta \mathcal{J}(\theta) = \nabla_\theta \left( \sum_{k=1}^{K} \frac{|\mathcal{D}_k|}{|\mathcal{D}|} \mathbb{E} \left[ \frac{1}{G} \sum_{i=1}^{G} \frac{1}{|y_i|} \sum_{t=1}^{|y_i|} f_{i,t}(r_{i,t}(\theta)) \hat{A}_{i,k} \right] \right) \tag{17}$$

Since the advantage estimate $\hat{A}_{i,k}$ is treated as a constant (stop-gradient) during the RL training process, the gradient operator acts solely on the $f_{i,t}$ term:

$$= \sum_{k=1}^{K} \frac{|\mathcal{D}_k|}{|\mathcal{D}|} \mathbb{E} \left[ \frac{1}{G} \sum_{i=1}^{G} \frac{1}{|y_i|} \sum_{t=1}^{|y_i|} \nabla_\theta \left( f_{i,t}(r_{i,t}(\theta)) \right) \hat{A}_{i,k} \right] \tag{18}$$

Applying the chain rule $\nabla_\theta f(r) = f'(r) \nabla_\theta r$:

$$= \sum_{k=1}^{K} \frac{|\mathcal{D}_k|}{|\mathcal{D}|} \mathbb{E} \left[ \frac{1}{G} \sum_{i=1}^{G} \frac{1}{|y_i|} \sum_{t=1}^{|y_i|} f_{i,t}'(r_{i,t}) \cdot \nabla_\theta r_{i,t}(\theta) \cdot \hat{A}_{i,k} \right] \tag{19}$$

Next, we compute the gradient of $r_{i,t}$. Since the denominator $\pi_{\text{old}}$ is fixed, it is treated as a constant:

$$\nabla_\theta r_{i,t}(\theta) = \nabla_\theta \left( \frac{\pi_\theta(y_{i,t}|q, y_{i,<t})}{\pi_{\theta_{\text{old}}}(y_{i,t}|q, y_{i,<t})} \right) = \frac{1}{\pi_{\theta_{\text{old}}}(\dots)} \nabla_\theta \pi_\theta(y_{i,t}|q, y_{i,<t}) \tag{20}$$

Applying the log-derivative trick $\nabla x = x \cdot \nabla \log x$ (i.e., $\nabla \pi = \pi \nabla \log \pi$):

$$= \frac{1}{\pi_{\theta_{\text{old}}}(\dots)} \Big( \pi_\theta(y_{i,t}|q, y_{i,<t}) \cdot \nabla_\theta \log \pi_\theta(y_{i,t}|q, y_{i,<t}) \Big) \tag{21}$$

$$= \underbrace{\frac{\pi_\theta(y_{i,t}|q, y_{i,<t})}{\pi_{\theta_{\text{old}}}(y_{i,t}|q, y_{i,<t})}}_{r_{i,t}} \cdot \nabla_\theta \log \pi_\theta(y_{i,t}|q, y_{i,<t}) \tag{22}$$

Substituting $\nabla_\theta r_{i,t} = r_{i,t} \nabla_\theta \log \pi_\theta$ back into the original expression yields Eq. 2:

$$\nabla_\theta \mathcal{J}(\theta) = \sum_{k=1}^{K} \frac{|\mathcal{D}_k|}{|\mathcal{D}|} \mathbb{E} \left[ \frac{1}{G} \sum_{i=1}^{G} \frac{1}{|y_i|} \sum_{t=1}^{|y_i|} f'_{i,t}(r_{i,t}) r_{i,t} \nabla_\theta \log \pi_\theta(y_{i,t}|q, y_{i,<t}) \hat{A}_{i,k} \right] \tag{23}$$

## C.2. Derivation of Eq. 4

We seek to compute the gradient of the term $W_{i,t}(\theta)\hat{A}_{i,k}$ with respect to $\theta$. Using the product rule:

$$\frac{d}{d\theta} \left( W_{i,t}(\theta)\hat{A}_{i,k} \right) = \underbrace{\frac{dW_{i,t}(\theta)}{d\theta} \hat{A}_{i,k}}_{\text{Term I}} + \underbrace{W_{i,t}(\theta) \frac{d\hat{A}_{i,k}}{d\theta}}_{\text{Term II}} \tag{24}$$

Here, "Term I" corresponds to the first part of Eq. 4. The primary focus is to derive the gradient $\frac{d\hat{A}_{i,k}}{d\theta}$ within "Term II".

Employing the chain rule, we decompose the derivative of $\hat{A}_{i,k}$ with respect to $\theta$ into three components:

$$\frac{d\hat{A}_{i,k}}{d\theta} = \frac{\partial \hat{A}_{i,k}}{\partial A_{i,k}} \cdot \frac{\partial A_{i,k}}{\partial x_{i,k}} \cdot \frac{\partial x_{i,k}}{\partial \theta} \tag{25}$$

The definitions of the latter two terms are given by: $\frac{\partial A_{i,k}}{\partial x_{i,k}} = g'_k(x_{i,k})$ and $\frac{\partial x_{i,k}}{\partial \theta} = H'_k(\theta, q, t)$. The core challenge lies in deriving the first term, $\frac{\partial \hat{A}_{i,k}}{\partial A_{i,k}}$.

Recall the definition: $\hat{A}_{i,k} = \frac{A_{i,k} - \mu}{\sigma}$, where $\mu = \frac{1}{G} \sum_{j=1}^{G} A_{j,k}$ and $\sigma = \sqrt{\frac{1}{G} \sum_{j=1}^{G} (A_{j,k} - \mu)^2}$. Note that both $\mu$ and $\sigma$ are functions of $A_{i,k}$.

We first compute the auxiliary derivatives:

$$\frac{\partial \mu}{\partial A_{i,k}} = \frac{1}{G}, \quad \frac{\partial \sigma}{\partial A_{i,k}} = \frac{1}{G} \frac{A_{i,k} - \mu}{\sigma} = \frac{1}{G} \hat{A}_{i,k} \tag{26}$$

Applying the quotient rule to compute $\frac{\partial \hat{A}_{i,k}}{\partial A_{i,k}}$:

$$\frac{\partial \hat{A}_{i,k}}{\partial A_{i,k}} = \frac{\partial}{\partial A_{i,k}} \left( \frac{A_{i,k} - \mu}{\sigma} \right) \tag{27}$$

$$= \frac{(1 - \frac{\partial \mu}{\partial A_{i,k}})\sigma - (A_{i,k} - \mu)\frac{\partial \sigma}{\partial A_{i,k}}}{\sigma^2} \tag{28}$$

$$= \frac{(1 - \frac{1}{G})\sigma - (A_{i,k} - \mu)(\frac{1}{G}\hat{A}_{i,k})}{\sigma^2} \tag{29}$$

$$= \frac{\frac{G-1}{G}\sigma - (\sigma \hat{A}_{i,k})\frac{\hat{A}_{i,k}}{G}}{\sigma^2} \quad \text{(substituting } A_{i,k} - \mu = \sigma \hat{A}_{i,k}) \tag{30}$$

$$= \frac{1}{G\sigma} \left[ (G-1) - \hat{A}_{i,k}^2 \right] \tag{31}$$

Substituting the above results back into the chain rule formulation:

$$\frac{d\hat{A}_{i,k}}{d\theta} = \underbrace{\frac{1}{G\sigma} \left[ (G-1) - \hat{A}_{i,k}^2 \right]}_{\partial \hat{A}/\partial A} \cdot \underbrace{g_k'(H_k(\theta, q, t))}_{\partial A/\partial x} \cdot \underbrace{H_k'(\theta, q, t)}_{\partial x/\partial \theta} \tag{32}$$

Finally, substituting this result into "Term II" and rearranging yields:

$$\frac{d}{d\theta} \left( W_{i,t}(\theta)\hat{A}_{i,k} \right) = W_{i,t}'(\theta)\hat{A}_{i,k} + W_{i,t}(\theta) \times \frac{g_k'(H_k(\theta, q, t))}{G\sigma} \left[ (G-1) - \hat{A}_{i,k}^2 \right] H_k'(\theta, q, t) \tag{33}$$

**Remarks on Mathematical Subtleties.** We address two crucial mathematical nuances in the derivations above:

**1. The Dual Role of Advantage $\hat{A}_{i,k}$.** It is crucial to distinguish the mathematical treatment of the advantage term between the gradient computation (Eq. 2) and the dynamics analysis (Eq. 4). In the actual RL optimization step (Eq. 1 → Eq. 2), $\hat{A}_{i,k}$ is an observed scalar computed from sampled trajectories. The optimization objective treats these advantages as fixed "critic" signals to guide the policy, hence the stop-gradient operation. However, Eq. 4 performs a theoretical second-order analysis to examine the *training dynamics*—specifically, how the gradient vector itself evolves as the model parameters $\theta$ shift. In this analytical context, the advantage is strictly a function of the model's capability: as $\theta$ updates, the distribution of responses changes, altering the raw metrics $x$ and the resulting advantages. Therefore, we must differentiate $\hat{A}_{i,k}$ with respect to $\theta$ to capture the sensitivity of the learning signal.

**2. The Direct Path Approximation in Chain Rule.** In Eq. 25, we express the total derivative $\frac{d\hat{A}_{i,k}}{d\theta}$ but focus predominantly on the partial derivative chain through the $i$-th sample itself: $\frac{\partial \hat{A}_{i,k}}{\partial A_{i,k}} \frac{\partial A_{i,k}}{\partial \theta}$. Technically, since the normalization statistics $\mu$ and $\sigma$ depend on the entire group $\{A_{j,k}\}_{j=1}^G$, a full Jacobian expansion would include cross-terms $\sum_{j \neq i} \frac{\partial \hat{A}_{i,k}}{\partial A_{j,k}} \frac{\partial A_{j,k}}{\partial \theta}$. We adopt this *Direct Path Approximation* for two reasons: (1) **Dominance:** The self-influence term $(\partial \hat{A}_{i,k}/\partial A_{i,k})$ typically dominates the gradient magnitude compared to the cross-terms, which are suppressed by a factor of $1/G$ via the mean and variance. (2) **Physical Interpretability:** Our objective is to isolate the mechanism of Reward Shaping—how improving a specific response $y_i$ amplifies its own learning signal. Treating the group statistics $(\mu, \sigma)$ as a quasi-static baseline allows us to cleanly demonstrate the "Matthew Effect" without the obfuscation of complex cross-sample interference. For a rigorous derivation including the full Jacobian and cross-terms, please refer to Appendix G.

## D. Discussion on the Differentiability of Reward Mapping Functions

The derivations presented in the main text (Eqs. 3 and 4) are predicated on the assumption that the reward mapping function $g_k(\cdot)$ possesses well-behaved differentiability. However, in practical applications, scenarios exist where $g_k(\cdot)$ is not fully differentiable. We first classify common non-differentiable scenarios and then propose respective solutions.

### D.1. Classification of Non-Smooth Scenarios

Based on the nature of the task metric $x_{i,k}$, real-world scenarios can be categorized into two types:

*Case I: Discrete Metric.* This is a typical setting for tasks such as mathematical reasoning or multiple-choice questions. Here, $x_{i,k}$ takes values from a discrete set (e.g., $\{0,1\}$).

$$x_{i,k} \in \{0,1\}, \quad A_{i,k} = g_k(x_{i,k}) = x_{i,k} \tag{34}$$

In this case, not only might $g_k$ be step-like, but more critically, the variation of $x_{i,k}$ with respect to the model parameters $\theta$ is discontinuous.

*Case II: Continuous Metric with Discrete Mapping.* The underlying metric $x_{i,k}$ (e.g., IoU, BLEU) is dense in the real domain but is artificially mapped to a reward via a step function:

$$A_{i,k} = g_k(x_{i,k}) = \mathbb{I}(x_{i,k} \geq \tau) = \begin{cases} 1, & \text{if } x_{i,k} \geq \tau, \\ 0, & \text{otherwise.} \end{cases} \tag{35}$$

where $\tau$ is a threshold.

## D.2. Analysis of Update Signal Failure

Although these two scenarios manifest differently, both lead to "signal loss" during the optimization process:

*For Case II:* Here, $x_{i,k}$ is sensitive to $\theta$; that is, a minor update in model parameters $\Delta\theta$ can induce a slight change in $x_{i,k}$ (e.g., IoU improving from 0.45 to 0.48). However, recalling the key term in Eq. 4:

$$\frac{d}{d\theta} A_{i,k} = \underbrace{g'_k(x_{i,k})}_{=0} \cdot \underbrace{H'_k(\theta)}_{\neq 0} = 0$$

Since $g_k$ is a step function, its derivative $g'_k$ is zero everywhere except at the jump points. This directly causes a breakdown in the chain rule; although the model makes "marginal improvements" in capability, it fails to receive positive gradient feedback.

*For Case I:* Since $x_{i,k}$ itself undergoes discrete jumps, perturbations in the model parameters $\theta$ are often insufficient to alter the value of $x_{i,k}$ (e.g., logits change but the `argmax` result remains unchanged). This implies that even if the model improves, it does not receive the due positive feedback because the outcome remains static.

## D.3. Unified Solution and Verification

To address the aforementioned "signal loss" and leverage the non-linear reward amplification mechanism proposed in the main text, we adopt specific strategies for each scenario, unifying them under a differentiable optimization framework.

**Optimization for Case II: Smoothing Reward Mapping** For scenarios where the metric $x_{i,k}$ is dense, the solution is to replace the step function $g_k$ with a smooth non-linear function. This aligns precisely with the core proposition of the main text. To verify this, we conducted comparative experiments on an image tampering localization task (based on the IoU metric). As shown in Table 7, compared to using step-based rewards, the exponential reward function $g_k(x) = e^{ax}$ designed in the main text achieves superior performance.

*Table 7.* Comparison of different reward functions on the image tampering localization task. The exponential reward (proposed) outperforms the step-based reward in terms of IoU.

| Reward Function | Parameter | IoU (%) |
|---|---|---|
| Step-based Reward | $\tau = 0.6$ | 51.29 |
| Exponential Reward (Ours) | $a = 3$ | **54.26** |

**Optimization for Case I: Metric Continuous Relaxation** For scenarios where $x_{i,k}$ is discrete, direct optimization is infeasible, necessitating the introduction of a continuous proxy variable. We extend the original metric as follows:

$$\tilde{x}_{i,k} = x_{i,k} + \lambda x'_{i,k} \tag{36}$$

where $x'_{i,k}$ is a continuous variable reflecting the quality of the model's inference process (e.g., the Perplexity of the reasoning chain or the confidence of the logits), and $\lambda$ is a weighting coefficient. At this point, although $x_{i,k}$ remains discrete, the combined metric $\tilde{x}_{i,k}$ becomes a dense variable. This successfully transforms Case I into Case II. Consequently, we can apply the conclusions from the main text to design a reward function for $\tilde{x}_{i,k}$ with properties of the form $g_k(\cdot) = e^{ax}$. Thus, even if the final answer $x_{i,k}$ has not yet flipped, the model can still receive gradient signals via $x'_{i,k}$, driving continuous evolution in the correct direction under the non-linear reward mechanism described in the main text.

## E. Clarification on the Differentiability and Definition of Task Capability Metric $H_k$

In Sec. 4.1, we introduced the term $H_k(\theta, q, \tau)$ to model the task-specific performance metric dependent on model parameters $\theta$. We acknowledge that in the practical implementation of RL, the generation process involves discrete sampling $y \sim \pi_\theta$, making individual sample metrics (e.g., IoU, Accuracy) non-differentiable with respect to $\theta$ in the standard computational graph.

To ensure the rigorousness of our theoretical analysis in Eq. 4, we formally define $H_k(\theta, q, \tau)$ as the **Expected Capability** of the model for a specific task query $q$:

$$H_k(\theta, q, \tau) = \mathbb{E}_{y \sim \pi_\theta(\cdot|q)} [M_k(y, q)] \tag{37}$$

where $M_k(y, q)$ is the raw discrete metric (e.g., IoU or Binary Indicator) for a generated response $y$.

**Justification for Differentiability:** While distinct realizations $y_i$ are discrete, the expectation $H_k(\theta, q, \tau)$ is a smooth, continuous function of the policy parameters $\theta$ (assuming the policy $\pi_\theta$ is differentiable, which is true for Softmax-based MLLMs). Therefore, the derivative $H'_k(\theta) = \nabla_\theta H_k(\theta, q, \tau)$ is mathematically well-defined. It represents the sensitivity of the model's average performance to parameter updates. This is consistent with the Policy Gradient Theorem, where gradients estimate the direction that maximizes this expected return.

**Interpretation of Eq. (4) under this Definition:** The analysis in Eq. (4) investigates the *training dynamics* of the gradient vector field.

- **Low-Performance Plateau ($H'_k \approx 0$):** For high-difficulty tasks, the probability mass assigned to high-quality responses is often negligible and located in "flat" regions of the probability landscape. Consequently, the gradient of the expected performance, $\nabla_\theta \mathbb{E}[M_k]$, vanishes (i.e., marginal changes in $\theta$ do not statistically improve the metric).

- **Role of Eq. (4):** The equation serves as a continuous analytical approximation to demonstrate how the *magnitude* of the update signal is suppressed by this plateau effect. By analyzing the second-order dynamics of this expectation, we justify why a steeper reward mapping $g_k(\cdot)$ is necessary to counteract the vanishing sensitivity of the expected capability.

Thus, the term $\frac{\partial x_{i,k}}{\partial \theta}$ in our derivation should be interpreted as differentiating the continuous latent capability surface approximated by the expectation, rather than a discrete sampling operation.

## F. Detailed Analysis of Gradient Dynamics Components

In the main text, we derived the second-order derivative of the objective function (Eq. 4) to analyze the training dynamics. Here, we provide a microscopic interpretation of this equation by decomposing the second term into three distinct physical components. Recall the core term governing the acceleration of the gradient updates:

$$\mathcal{A}_{i,k} = \underbrace{\frac{g'_k(x_{i,k})}{G\sigma}}_{\mathcal{C}_{\text{map}}} \times \underbrace{\left[(G-1) - \hat{A}^2_{i,k}\right]}_{\mathcal{C}_{\text{stat}}} \times \underbrace{H'_k(\theta)}_{\mathcal{C}_{\text{task}}} \tag{38}$$

This decomposition reveals the interaction between reward design, statistical grouping, and task difficulty. We analyze each component below.

### F.1. Component I: The Reward-Variance Leverage ratio ($\mathcal{C}_{\textbf{map}}$)

The term $\mathcal{C}_{\text{map}} = \frac{g'_k(x_{i,k})}{G\sigma}$ represents the *sensitivity of the reward signal normalized by the group diversity*. A critical question arises: simply increasing the scale of rewards (e.g., multiplying by a constant) increases both the derivative $g'_k(\cdot)$ and the standard deviation $\sigma$ proportionally, resulting in no net change to $\mathcal{C}_{\text{map}}$. Why, then, do we advocate for a steeper, convex reward mapping?

**Mathematical Explanation:** Consider two mapping functions for a set of task metrics $\{x_j\}_{j=1}^G$:

- **Linear Mapping** ($g(x) = \alpha x$): The derivative is constant $g'(x) = \alpha$. The standard deviation scales as $\sigma_g = \alpha\sigma_x$. The ratio becomes $\frac{\alpha}{G\alpha\sigma_x} \propto \frac{1}{\sigma_x}$. Here, the gradient contribution is uniform across all samples and independent of the absolute performance level $x$.

- **Convex Mapping** (e.g., $g(x) = e^{\beta x}$): The derivative is $g'(x) = \beta e^{\beta x} = \beta g(x)$, which depends on the value of $x$ itself. The standard deviation $\sigma$ represents the *average* spread of the group.

For a high-performing response (where $x_{i,k}$ is the maximum in the group), a convex mapping ensures that the local slope $g'(x_{i,k})$ grows significantly faster than the group standard deviation $\sigma$. Mathematically, if $x_{i,k} > x_{j,k}$ for $j \neq i$, convexity implies:

$$\frac{g'(x_{i,k})}{\sigma(g(x))} \gg \frac{g'(x_{j,k})}{\sigma(g(x))} \tag{39}$$

By designing $g_k(\cdot)$ with a progressively steepening slope, we maximize the ratio $\mathcal{C}_{\text{map}}$ specifically for superior responses. This effectively creates a "signal amplifier" that functions only when the model generates high-quality outputs, counteracting the suppression from other terms.

### F.2. Component II: The Statistical Stability Factor ($\mathcal{C}_{\textbf{stat}}$)

The term $\mathcal{C}_{\text{stat}} = (G-1) - \hat{A}_{i,k}^2$ originates from the differentiation of the Z-score normalization.

**Independence from Task Difficulty:** It is crucial to note that this term is **statistically invariant** to task difficulty. Regardless of whether a task is easy (all raw scores $x \approx 0.9$) or hard (all raw scores $x \approx 0.1$), the normalized advantages $\hat{A}_{i,k}$ follow a distribution with zero mean and unit variance (approximately $\mathcal{N}(0,1)$ for large $G$). Consequently, the magnitude of $\hat{A}_{i,k}^2$ is determined solely by the *relative ranking* of response $i$ within its group, not by the absolute difficulty of the task.

**Physical Meaning:** This term acts as a stabilizer. When a response is an outlier (i.e., $|\hat{A}_{i,k}|$ is very large, making $\hat{A}_{i,k}^2 > G-1$), $\mathcal{C}_{\text{stat}}$ can become negative, effectively reversing the "acceleration" direction to prevent overfitting to extreme samples. However, for most samples, this term provides a constant baseline scaling roughly proportional to the group size $G$, treating simple and complex tasks equally.

### F.3. Component III: The Task Difficulty Sensitivity ($\mathcal{C}_{\textbf{task}}$)

As discussed in the main text, $\mathcal{C}_{\text{task}} = H'_k(\theta)$ reflects the marginal performance gain per parameter update. This is the primary source of the "Matthew Effect": for complex tasks, $H'_k \to 0$ (the performance plateau), which nullifies the entire gradient update $\mathcal{A}_{i,k}$ regardless of the other terms.

**Summary of Interaction:** Since $\mathcal{C}_{\text{stat}}$ is task-agnostic and $\mathcal{C}_{\text{task}}$ vanishes for hard tasks, the only mechanism to restore gradients for complex tasks is to significantly amplify $\mathcal{C}_{\text{map}}$. Our nonlinear reward shaping achieves this by ensuring that when a rare high-quality response appears in a hard task, its corresponding ratio $\frac{g'}{\sigma}$ is sufficiently large to resurrect the vanishing gradient.

## G. Analysis of Gradient Dynamics under Total Derivative

In the main text (Sec. 4.1) and Appendix C.2, we derived the gradient sensitivity term $\frac{\partial \hat{A}_{i,k}}{\partial A_{i,k}}$ by focusing on the *self-sensitivity* of a response's advantage with respect to its own raw reward. Here, we extend this analysis to the *Total Derivative* scenario, considering the coupled influence where a change in model parameters $\theta$ simultaneously affects all responses $\{y_j\}_{j=1}^G$ in the group.

### G.1. Total Derivative Derivation

Let the set of raw rewards in a group be $\mathcal{A} = \{A_{1,k}, \ldots, A_{G,k}\}$. The normalized advantage for the $i$-th response is $\hat{A}_{i,k} = \frac{A_{i,k} - \mu}{\sigma}$, where $\mu(\mathcal{A})$ and $\sigma(\mathcal{A})$ depend on all rewards in the group.

The total derivative of $\hat{A}_{i,k}$ with respect to the model parameters $\theta$ is given by the chain rule, summing over the contributions from all samples $j \in \{1, \ldots, G\}$:

$$\frac{d\hat{A}_{i,k}}{d\theta} = \sum_{j=1}^{G} \frac{\partial \hat{A}_{i,k}}{\partial A_{j,k}} \frac{\partial A_{j,k}}{\partial \theta} \tag{40}$$

Recall from Appendix C.2 that the partial derivatives are:

- **Self-Sensitivity** ($j = i$):

$$\frac{\partial \hat{A}_{i,k}}{\partial A_{i,k}} = \frac{1}{G\sigma} \left[ (G-1) - \hat{A}_{i,k}^2 \right] \tag{41}$$

- **Cross-Sensitivity** ($j \neq i$): Using the derivatives $\frac{\partial \mu}{\partial A_{j,k}} = \frac{1}{G}$ and $\frac{\partial \sigma}{\partial A_{j,k}} = \frac{\hat{A}_{j,k}}{G}$, we derive:

$$\begin{aligned}
\frac{\partial \hat{A}_{i,k}}{\partial A_{j,k}} &= \frac{\sigma(0 - \frac{\partial \mu}{\partial A_{j,k}}) - (A_{i,k} - \mu)\frac{\partial \sigma}{\partial A_{j,k}}}{\sigma^2} \\
&= \frac{-\frac{\sigma}{G} - (\sigma \hat{A}_{i,k})\frac{\hat{A}_{j,k}}{G}}{\sigma^2} \\
&= -\frac{1}{G\sigma} \left[ 1 + \hat{A}_{i,k} \hat{A}_{j,k} \right]
\end{aligned} \tag{42}$$

Substituting these into Eq. 40, and denoting the raw reward gradient as $A'_j = \frac{\partial A_{j,k}}{\partial \theta} = g'_k(x_{j,k})H'_k(\theta)$, we obtain the full expression:

$$\frac{d\hat{A}_{i,k}}{d\theta} = \underbrace{\frac{1}{G\sigma}[(G-1) - \hat{A}_{i,k}^2]A'_i}_{\text{Self-Term}} - \underbrace{\frac{1}{G\sigma}\sum_{j\neq i}(1 + \hat{A}_{i,k}\hat{A}_{j,k})A'_j}_{\text{Cross-Term}} \tag{43}$$

### G.2. Analysis of Self-Term Dominance in High-Performance Regions

Equation 43 decomposes the total gradient into a *Self-Term* (driven by the sample's own raw reward gradient $A'_i$) and a *Cross-Term* (driven by the gradients of other samples $A'_j$). Here, we demonstrate that for a high-quality response $i$ under a steep convex reward mapping (e.g., $g_k(x) = e^{\alpha x}$ with $\alpha \gg 1$), the **Self-Term determines the gradient direction**, rendering the Cross-Term negligible.

**1. Gradient Disparity under Steep Mapping.** Consider a group where the $i$-th response is superior ($x_{i,k} > x_{j,k}$ for $j \neq i$). With the exponential reward $A = e^{\alpha x}$, the sensitivity of the raw reward to model parameters is:

$$A'_k = \frac{\partial A_k}{\partial \theta} \propto \alpha e^{\alpha x_k}. \tag{44}$$

Due to the exponential property, the ratio of gradients between the superior response and inferior responses grows exponentially with the performance gap:

$$\frac{\|A'_i\|}{\|A'_j\|} \approx e^{\alpha(x_i - x_j)}. \tag{45}$$

For a sufficiently large $\alpha$ (e.g., $\alpha = 3$), even a moderate performance advantage results in $A'_i$ being orders of magnitude larger than $A'_j$.

**2. Dominance in the Total Derivative.**  We examine the total derivative for the superior sample $i$ (Eq. 43):

$$\frac{d\hat{A}_{i,k}}{d\theta} = \underbrace{C_{\text{self}} \cdot A'_i}_{\text{Self-Term}} - \underbrace{\frac{1}{G\sigma} \sum_{j \neq i} (1 + \hat{A}_{i,k}\hat{A}_{j,k}) A'_j}_{\text{Cross-Term}} \tag{46}$$

where $C_{\text{self}} = \frac{1}{G\sigma}[(G-1) - \hat{A}_{i,k}^2]$ is a bounded statistical coefficient.

Although the normalization term $\sigma$ scales with the reward magnitude (preventing numerical explosion), the *relative contribution* of the two terms is dictated by the raw gradients $A'$.

- The Self-Term is driven by $A'_i$ (the dominant signal).

- The Cross-Term is a weighted sum of $A'_j$ for $j \neq i$. Since $x_j < x_i$, these $A'_j$ terms are exponentially suppressed (vanishingly small).

Mathematically, as long as the statistical coefficient $C_{\text{self}}$ is not exactly zero (which only occurs in the trivial limit of a single outlier dominating variance entirely), the total derivative is approximated by:

$$\frac{d\hat{A}_{i,k}}{d\theta} \approx C_{\text{self}} \cdot A'_i + O(e^{-\alpha\Delta x}) \approx \text{Self-Term}. \tag{47}$$

**3. Physical Interpretation: Decoupled Optimization.**  This result provides a crucial physical insight: Superior samples are effectively decoupled from the group noise. While the *magnitude* of the update is normalized by the group statistics (via $\sigma$), the *direction* of the update for a high-performing path is determined almost exclusively by its own gradient structure. The "drag" effects from poorer samples (the Cross-Terms) are filtered out by the steep reward mapping because those samples have near-zero gradients.

**Conclusion.**  This analysis justifies the *Direct Path Approximation*. In the high-performance regime, the complex coupled derivatives simplify: the model learns to reinforce its best paths based on their own merits, undisturbed by the gradient noise from inferior parallel samples.

# H. Details on Filtering and Verification Criteria

### H.1. Filtering

To construct a high-quality training set, we established strict filtering thresholds tailored to the specific characteristics of each task. A sample is retained in $D_s$ only if the CoT-based prediction of the model meets the following criteria:

**Binary Classification:** Only samples where the prediction matches the Ground Truth exactly (Correct Prediction) are retained.

**Image Tampering Localization:** Samples are retained if the Intersection over Union (IoU) between the predicted region and the ground truth satisfies IoU $\geq 0.75$.

**Text Tampering Localization:** Samples are retained if the F1 Score of the predicted text segment against the ground truth satisfies F1 Score $\geq 0.75$.

**Video Temporal Localization:** Samples are retained if the temporal Intersection over Union (tIoU) between the predicted time segment and the ground truth satisfies tIoU $\geq 0.75$.

### H.2. Verification

To ensure the high quality and reliability of the generated Chain-of-Thought (CoT), we implemented an evaluation mechanism based on state-of-the-art (SOTA) MLLMs across the *Seed Bootstrapping*, *Iterative Self-Evolution*, and *De-biased Hard Synthesis* stages.

**Evaluation Metric System.** We defined core metrics across distinct dimensions for the screening process. Only samples that simultaneously satisfy all the following criteria are included in the dataset $\mathcal{D}$:

1. *Content Faithfulness:* The reasoning steps within the CoT must be strictly grounded in the content of the image, video, or text. Any reasoning path involving non-existent objects, incorrect descriptions of color or location, or "hallucinations" is deemed unqualified.

2. *Logical Coherence & Consistency:*

   - *Closed-Loop Deduction:* The reasoning process must flow logically toward the final answer, avoiding disjointed statements or abrupt jumps to conclusions.
   - *Absence of "Hindsight Bias":* The reasoning process should simulate a progressive human observation process. It is strictly prohibited to reverse-engineer the reasoning from the known answer in early steps (this criterion is enforced with particular rigor during the De-biased Hard Synthesis stage).

**SOTA MLLM Verification and Generation Prompts:** We employ models from $\mathcal{M} = \{$Seed1.6-VL, Gemini3, ChatGPT5$\}$ to serve as "Judges" and "Refiners." The core prompt templates used in each stage are detailed below.

*General CoT Quality Verification*

During the iterative process, we use the following prompt to instruct the SOTA MLLM to act as a judge, performing a binary classification (Pass/Fail) on the logical quality of the generated data.

---

**Prompt for CoT Verification (Judge)**

**System Instruction:** You are an expert AI assistant tasked with evaluating the quality of reasoning chains (CoT) generated by Multimodal LLMs. You will be provided with Image/Video/Text, a Question, the Ground Truth Answer, and a Candidate CoT.
**Your Task:** Verify the Candidate CoT based on the following strict criteria: 1. *Content Faithfulness:* Are all descriptions in the reasoning steps actually present in the image/video/text? Reject any hallucinations. 2. *Logic Flow:* Is the reasoning step-by-step and logically sound? There should be no logical gaps.
**Input Data:** [Image/Video/Text] **Question:** {Task-based Question} **Ground Truth:** {GT} **Candidate CoT:** {Generated_CoT}
**Output Format:** Return strictly in JSON format: { "judgment": "Pass" or "Fail", "reason": "Brief explanation of the failure or confirmation of quality." }

---

*De-biased Refinement (Used in Collaborative Hard-CoT Synthesis)*

For hard samples, CoTs generated with access to the Ground Truth (GT) often exhibit "hindsight bias." We use the following prompt to drive the "Refiner" model to eliminate these "prophetic" characteristics, simulating a genuine exploratory reasoning process.

---

**Prompt for De-biased Refinement (Refiner)**

**System Instruction:** You are a logic refinement expert. You will be given a Question, an Image/Video/Text, and a "Raw CoT". The Raw CoT reaches the correct answer but was generated with GT of the solution, making it sound artificial or backward-reasoned.
**Your Task:** Rewrite the CoT to simulate a human viewing the image for the first time. 1. *Remove Hindsight:* Eliminate phrases like "Since we know the answer is..." or "To match the label...". 2. *Add Exploratory Steps:* Include the natural process of scanning the image, identifying key features, and linking them to the question. 3. *Keep Logic Intact:* Ensure the reasoning still rigorously leads to the correct answer.
**Input Data:** [Image/Video/Text] **Question:** {Question} **Raw CoT:** {Initial_CoT_with_GT}
**Output:** Provide ONLY the refined CoT.

---

## H.3. Human Validation

To further validate the reliability of our automated pipeline and ensure the absence of model-specific biases or hallucinations, we conducted a rigorous human evaluation. We randomly sampled 300 instances from the data pool that passed the automated *Verification* stage and 300 instances from the output of the *De-biased Refinement* stage.

Human experts reviewed these samples based on the strict content faithfulness and logical coherence criteria defined above. The inspection results are as follows:

- **Verification Stage:** Among the 300 randomly selected samples that passed the automated judge, only 2 samples were identified as unqualified by human annotators (an error rate of $\approx 0.67\%$). This indicates a high agreement rate between our automated verifier and human judgment.

- **Refinement Stage:** Among the 300 randomly selected samples processed by the Refiner, only 5 samples failed to meet the standards during human inspection (an error rate of $\approx 1.67\%$). This confirms that the refinement process effectively eliminates hindsight bias while maintaining logical integrity.

These results substantiate the high quality and reliability of the constructed FSFR dataset.

## I. Analysis of Hindsight Bias in CoT Generation

To demonstrate the necessity of our *Self-Evolving Forensic CoT Generation* pipeline, we compared it against a baseline approach involving direct Ground Truth (GT) injection. In this baseline setting, we utilized the same set of SOTA MLLMs (as defined in Sec. 3.2) to generate Chain-of-Thought (CoT) annotations by explicitly providing the ground truth labels in the prompt. To ensure a fair comparison, we generated a dataset of identical size to $\text{FSFR}_{\text{sft}}$ and employed the exact same training recipe (SFT cold-start followed by ARSPO) for both models.

*Table 8.* Comparison between our Self-Evolving strategy and Direct GT Injection. The results highlight that relying on GT-guided CoT leads to suboptimal performance due to hindsight bias, particularly in reasoning-intensive localization tasks.

| CoT Generation Method | Binary Classification (ACC) | | | | Localization | | |
|---|---|---|---|---|---|---|---|
| | Text | Image | Video | Text-Image | Image (IoU) | Text (F1) | Video (tIoU) |
| Direct GT Injection | 92.15 | 84.47 | 91.30 | 69.12 | 51.50 | 52.84 | 46.10 |
| **Self-Evolving (Ours)** | **96.20** | **93.12** | **98.58** | **75.52** | **54.26** | **63.78** | **59.22** |
| *Performance Gap ($\Delta$)* | *-4.05* | *-8.65* | *-7.28* | *-6.40* | *-2.76* | *-10.94* | *-13.12* |

As presented in Table 8, the model trained with CoT generated via Direct GT Injection significantly underperforms our method across all tasks. The degradation is particularly severe in fine-grained tasks, such as Video Localization (tIoU drops by over 13%) and Text Localization (F1 score drops by nearly 11%).

We attribute this phenomenon to the quality of the reasoning paths. While injecting Ground Truth (GT) to force-guide generation ensures label correctness, it introduces *Hindsight Bias*, where models rationalize answers backward rather than performing genuine deduction. Such reasoning paths often lack the exploratory logic required for forensics (e.g., they skip critical evidence gathering). Consequently, using such biased data for SFT cold-start results in a policy that mimics superficial rationalization, thereby severely impairing the efficiency of subsequent RL exploration and optimization.

## J. Generalizability Analysis on Different Backbones

To further verify the unbiased nature of our high-quality cold-start dataset ($\text{FSFR}_{\text{sft}}$) and the universal applicability of the ARSPO optimization strategy, we conducted additional experiments using **InternVL3.5-8B** as the backbone model. This setup mirrors our main experiments: we first performed SFT using $\text{FSFR}_{\text{sft}}$, followed by reinforcement learning using the $\text{FSFR}_{\text{rl}}$ dataset with the ARSPO.

### J.1. Implementation Details and Results

The training hyperparameters (e.g., learning rate, batch size, and reward mapping slopes $a = 3$) remained consistent with those used for Qwen3VL-8B to ensure a fair evaluation of the method's robustness.

As shown in Table 9, applying ARSPO to InternVL3.5-8B yields substantial improvements across all tasks compared to the SFT baseline. Specifically, we observe a **+24.1%** gain in Video Temporal Localization (tIoU) and a **+17.8%** gain in Text Tampering Localization (F1).

Table 9. Performance comparison of ARSPO on the InternVL3.5-8B backbone. The results demonstrate that ARSPO consistently unlocks fine-grained reasoning capabilities, independent of the specific model architecture.

| Method | Bin. Cls. (ACC) | Image Loc. (IoU) | Text Loc. (F1) | Video Loc. (tIoU) |
|---|---|---|---|---|
| InternVL3.5 (SFT) | 52.91 | 49.50 | 43.10 | 31.50 |
| InternVL3.5 (ARSPO) | **89.37** | **53.60** | **60.90** | **55.60** |
| *Improvement (Δ)* | *+36.46* | *+4.10* | *+17.80* | *+24.10* |

## J.2. Discussion on Model Agnosticism and Unbiasedness

We note that the absolute performance of InternVL3.5-8B with ARSPO is slightly lower than that of Qwen3VL-8B (e.g., Video Loc: 55.60% vs. 59.22%). This performance gap is attributable to the inherent architectural and pre-training differences between the backbones. However, these results highlight two critical advantages of our framework:

**Data Unbiasedness via Cross-Architecture Transferability:** A primary concern with synthetic data generated by LLMs is the potential for "generator-specific bias"—where the data contains artifacts or hallucinations that only the source model (Qwen) can comprehend. However, the InternVL3.5 (SFT) baseline in Table 9 achieves competitive performance (e.g., 49.50% IoU in Image Loc.) solely through supervised training on FSFR$_{sft}$. This successful transfer to a heterogeneous architecture serves as strong empirical evidence that our dataset encapsulates generalizable forensic logic rather than model-specific noise. This unbiased nature is structurally guaranteed by our "Multi-Agent Collaborative Filtering and Evolution" pipeline (see Sec. 3.4), where distinct MLLMs were employed to filter out model-specific hallucinations during data construction.

**Methodological Longevity:** The consistent relative gains achieved by ARSPO (e.g., +36.46% in Binary Cls.) on a different architecture demonstrate that our optimization strategy serves as a fundamental enhancement layer. It is not dependent on the specific parameter distributions of Qwen3VL. This suggests that OmniVL-Guard remains valid and effective even as newer, more powerful backbones are introduced in the future.

## K. Hyperparameter for Dynamic Coefficient Adjustment Algorithm

Table 10. Hyperparameter settings for the Dynamic Coefficient Adjustment Algorithm.

| Symbol | Description | Value |
|---|---|---|
| ***Schedule & Initialization*** | | |
| $T_{warm}$ | Warm-up phase duration (steps) | 800 |
| $T$ | Coefficient update interval (steps) | 100 |
| ***Adjustment Dynamics*** | | |
| $\alpha_{boost}$ | Coefficient boost factor | 1.1 |
| $\alpha_{decay}$ | Coefficient decay factor | 0.9 |
| ***Thresholds*** | | |
| $\epsilon_{mom}$ | Momentum protection threshold | 0.02 |
| $\epsilon_{rescue}$ | Regression rescue threshold | 0.10 |
| | High-perf. threshold (Binary Cls.) | 0.10 |
| $\tau_{high}$ | High-perf. threshold (Image Loc.) | 0.50 |
| | High-perf. threshold (Text & Video Loc.) | 0.60 |

# L. Benchmark Brief Introduction

To rigorously evaluate generalization, we select a set of representative benchmarks. The key characteristics of each dataset are summarized below:

**FakeNewsCorpus** contains millions of text-only news articles collected from 1,001 curated domains on OpenSources, enriched with content from NYTimes and WebHose, aimed at supporting deep learning methods for fake news detection.

**MCFEND** provides Chinese multimodal news instances gathered from multiple platforms. Each instance combines news content (text, images, and metadata) with social context signals such as posts, comments, emojis, user profiles, and engagement metrics, enabling comprehensive multimodal analysis.

**FakeClue** focuses exclusively on images, covering seven categories of real and synthetic content. Each image comes with natural language annotations describing category-specific visual artifacts for training and evaluation of synthetic image detection.

**LOKI** targets AI-generated content across multiple modalities: video, image, 3D, text, and audio. It provides detailed annotations to support both authenticity verification and fine-grained anomaly explanation.

**ForgeryNet** combines image and video data for face forgery research. The dataset spans classification and localization tasks, offering extensive annotations to study both detection and spatial–temporal forgery localization.

**GenVidBench** features video-only content generated by state-of-the-art AI models. Rich semantic labels include object categories, actions, and positions, with cross-source and cross-generator diversity designed into the dataset.

**DVF** is dedicated to video forgery detection. It contains real and synthetic clips produced via multiple diffusion-based techniques, covering diverse resolutions and durations, and built through automated pipelines from real sources and text-/image-to-video generation.

**SAMM** combines real and manipulated image–text pairs, incorporating face and name swaps as well as emotion edits. Detailed annotations highlight both altered image regions and associated text, supporting fine-grained multimodal forgery detection.

**MDSM** provides multimodal content for media manipulation studies, including image–text pairs. Manipulations cover Face Swap (FS), Face Attribute editing (FA), and Text Fabrication (TF), with precise alignment and annotations for both images and text.

**DGM4** addresses large-scale machine-generated manipulations over image–text pairs. It documents manipulations like Face Swap (FS), Face Attribute (FA), Text Swap (TS), and Text Attribute (TA), providing fine-grained labels for detection and grounding tasks.

**NewsCLIPpings** explores out-of-context (OOC) inconsistencies in news. The image–text pairs are authentic individually but deliberately mismatched to induce contextual errors, suitable for evaluating contextual reasoning in models.

**ISOT** is composed of text-only news articles, labeled as real or fake, aggregated from multiple sources to support text-based fake news classification.

**CASIA v2** specializes in image-only forgeries. It includes authentic and tampered images (copy-move and splice), with ground-truth masks that support both detection and localization research.

**MM-FakeBench** collects mixed-source image–text pairs for misinformation studies. Manipulations involve textual veracity distortion, visual veracity distortion, and cross-modal consistency changes, annotated with multiple fine-grained forgery types.

**FakeSV** contains Chinese short-video news with multiple modalities: videos, audio, text, and social context metadata. Manual annotations classify content as real, fake, or debunked, supporting research on multimodal short-video misinformation detection.

# M. Full-Spectrum Forensic Reasoning (FSFR)

<div>

**Case1-Text Forgery Detection**

**Question:** Please identify whether this caption is Real or Fake.

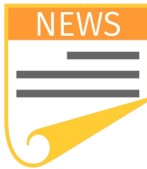 **Caption:** How Viola Davis's Experience with Childhood Hunger Inspired Her to Give Back.

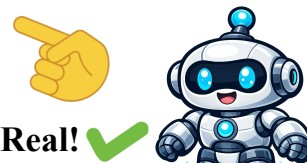

**Real!** ✅

**CoT:<think>** Got it, let's try to figure out if this news caption is real or fake. First, the caption says it's about Viola Davis's childhood hunger experience inspiring her to give back.

Wait, I remember that Viola Davis has been open about her childhood struggles, including poverty and hunger. Let me think—she's talked about growing up in Rhode Island with very little, sometimes not having enough food. And she does charitable work related to food insecurity, right? Like supporting organizations that help feed kids or families in need.

Is there any reason to think this is fake? The caption is a general statement about her experience inspiring giving back, which aligns with what's publicly known about her. There's no obvious misinformation here, no outlandish claim. So probably real.**</think>**

**<answer>Answer: Real.</answer>**

</div>

## Case2-Image Forgery Detection

**Question:**Please determine if the image is Real, AI-generated, or Tampered.

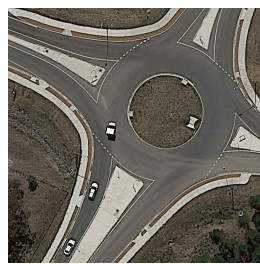 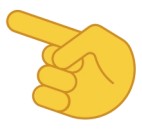 **Real!** ✅ 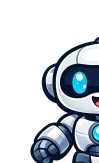

**CoT:<think>**Got it, let's try to figure out if this image is real, AI-generated, or tampered. First, I need to look at the details. The image is an aerial view of a roundabout. Let's start with the road markings: the lines are consistent, no weird distortions. The roundabout's center has grass that looks natural, not pixelated or odd. Then the cars—there are a few vehicles, their shapes are normal, no blurry or inconsistent parts. The surrounding areas: the edges of the roads have concrete curbs and some grass/dirt patches that look realistic, like real terrain.

Wait, are there any signs of AI generation? AI often makes mistakes in small details—like inconsistent lines, weird object shapes, or unnatural textures. Let's check the traffic lights or signs: there are some poles, but they look okay. The road surfaces have a uniform texture, no strange patterns. The way the roads curve around the roundabout is logical, like a real intersection design. Tampered? Is there any part that looks edited? The image seems cohesive, no obvious cuts or mismatched areas. All elements fit together naturally.

So putting it all together: the image has consistent, realistic details, no AI artifacts, no signs of tampering. So it's real.**</think>**

**<answer>Answer: Real.</answer>**

## Case3-Video Forgery Detection

**Question:** Please determine if the video is Real, AI-generated, or Tampered.

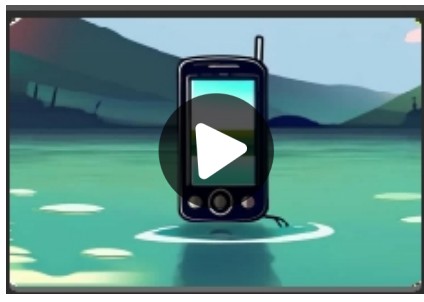

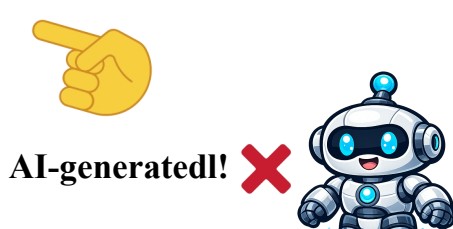

**AI-generatedl!** ❌

**CoT:<think>** Got it, let's try to figure out this. The video shows a phone standing upright in the middle of a body of water, right? First, real phones don't usually stand like that in water—they'd sink or float on their side, not upright with the screen showing. Also, the style looks like an animation, which isn't real footage.

The background is a cartoonish landscape with mountains and water that has a flat, illustrated look. So real videos are shot with cameras of real objects, but this is clearly an animated scene.

So it's AI-generated or maybe just animated, but according to the options, AI-generated makes sense here because the scenario is unrealistic (phone standing in water) and the visual style is not real. Let me confirm: real would be actual footage, tampered is modified real footage, but this is a created scene, so AI-generated.**</think>**

**<answer>Answer: AI-generated.</answer>**

## Case4-Text-Image Forgery Detection

**Question:**Please determine if the news is Real, AI-generated, or Tampered.

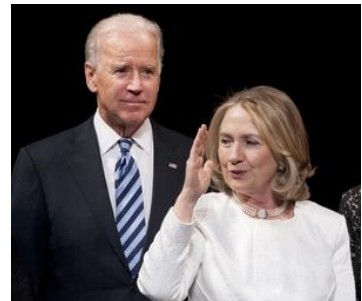

**Caption**:Former congresswoman Gabrielle Giffords and husband Mark Kelly.

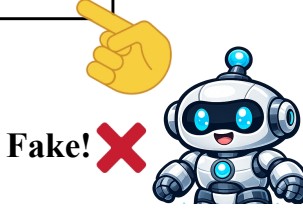

**Fake!** ❌

**CoT:<think>**Got it, let's try to figure this out. First, the image shows Joe Biden and Hillary Clinton, right? The caption says it's Gabrielle Giffords and her husband Mark Kelly. So the caption is definitely wrong for the image. The question says tampering could involve the image, caption, or both. Since the caption doesn't match the image, that's tampering.

Is it AI-generated? Probably not the image itself (it looks like a real photo of Biden and Clinton), but the pairing of the wrong caption with the image is tampering. So the answer should be Tampered.**</think>**

**<answer>Answer: Tampered.</answer>**

## Case5-Text Tampering Location

**Question:** Please locate the tampered words in the caption and provide the indices of the altered words as a list (starting from 0).

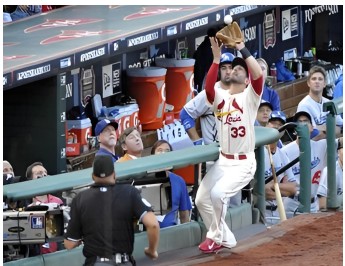

**Caption:** Game 2 Cardinals 1 Dodgers 0 Cardinals shortstop **Michael Dorn** catches a foul ball in the eighth inning Dodgers at Busch Stadium.

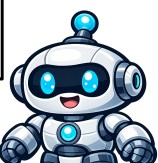

**CoT:<think>** Let's start by reading the caption to understand what it's describing. It says the Cardinals player catching the foul ball is shortstop Michael Dorn. Hmm, that seems specific. Now, looking at the image itself, I can see the player clearly—wearing a Cardinals jersey with number 33, red sleeves, white pants, and he's actively catching the ball near the dugout.

Wait a moment, does that name match up? I know Michael Dorn isn't the correct name for the Cardinals player who wears number 33 and would be making a play like this.

Let me double-check that—yes, the jersey number and the action don't align with that name at all. This confirms the caption is wrong here; the player isn't Michael Dorn. So, the inconsistency points to the words at indices [8,9] being manipulated.**</think>**

**<answer>Tampered words list: [8, 9]</answer>**

---

## Case6-Image Tampering Location

**Question:** Please locate the tampered region in the image.

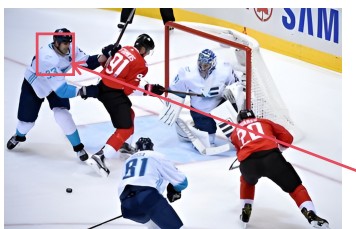

Caption: Team Canada players John Tavares and Steven Stamkos look for the puck between Team Europe players Marian Hossa and Ronnie Coleman.

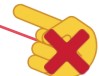

**CoT:<think>** Alright, let's look at this hockey scene. The caption says Ronnie Coleman is playing for Team Europe, which immediately throws me off—Ronnie Coleman is a famous bodybuilder, not a hockey player. That's a huge red flag, so the image is probably doctored. Now, scanning the players in white jerseys, my eyes go to number 33 on the left. Hmm, that face looks off. Let's check the lighting.

The rink has strong overhead lights, but the shadows on his face seem softer and angled differently compared to the crisp shadows from other players' gear. That doesn't match. Also, the skin texture—where's the sweat or frost you'd expect in a fast-paced game? It looks too clean, and the edges around the helmet blend weirdly, almost like a cut-and-paste job. And his expression... everyone else is intense, mid-action, but he seems oddly calm. This confirms it: that face isn't original. The manipulation is right there in that patch around his face. So, the tampered area is [161, 137, 196, 195].**</think>**

**<answer>Tampered bbox: {"boxes": [161, 137, 196, 195]}</answer>**

## Case7-Video Tampering Location

**Question:**Please identify the specific time segments that have been tampered with.

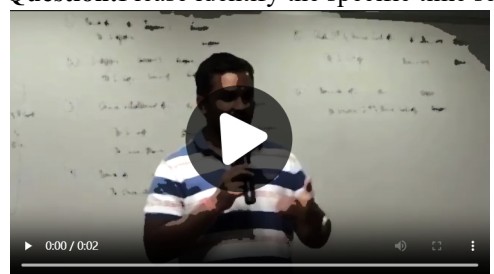 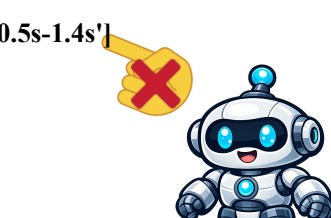 ['0.5s-1.4s']

**CoT:<think>**Alright, let's start by getting a feel for the overall video. The opening and closing parts seem normal—the subject's hand gestures are smooth and fluid, with natural motion blur that tracks the movement. That's my baseline.

But wait, focusing on the segment from around 0.5 seconds onward... something shifts. At exactly 0.5 seconds, the right hand jerks upward abruptly. That's odd—human gestures usually have a gradual acceleration, not this sudden jump. Hmm, that looks off compared to the earlier smooth motion.

Let's check further. Between 0.6 and 0.8 seconds, there's a frame where the fingers appear stretched and misaligned. That's a red flag; deepfakes often struggle with fine details like finger articulation. And then from 1.0 to 1.4 seconds, the motion blur becomes erratic—some frames have too much blur in the wrong direction, while others lack it entirely. That inconsistency suggests the synthesis algorithm failed here. After 1.4 seconds, everything returns to that natural baseline, which confirms the anomaly. So, the manipulated segment is clear: [0.5s-1.4s].**</think>**

**<answer>Tampered segment: ['0.5s-1.4s']</answer>**

# N. Inference Cases

## Case1-Text Forgery Detection

**Question:** Please identify whether this caption is Real or Fake.

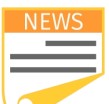 *Caption: Turning Bikes into Hybrids; Drooping Discount Lowers Biofuel Demand.* 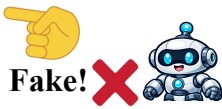

**Fake!**

**OmniVL-Guard:<think>** Got it, let's try to figure out if this news caption is real or fake. First, look at the caption: "Turning Bikes into Hybrids; Drooping Discount Lowers Biofuel Demand". Wait, "drooping discount"—that sounds really odd. Is there a term like that in economics or energy policy? Let me think. Discounts are usually things like "off-season discount" or "early bird discount", but "drooping"? That doesn't make sense. Also, the first part: turning bikes into hybrids—maybe electric bikes? But the second part's terminology is off. Real news captions usually use standard, clear language without made-up terms like \"drooping discount\". So this is probably fake because of the nonsensical term.**</think>**

**<answer>Answer: Fake.</answer>**

**Correct answer: Fake.**

## Case2-Image Forgery Detection

**Question:** Please determine if the image is Real, AI-generated, or Tampered.

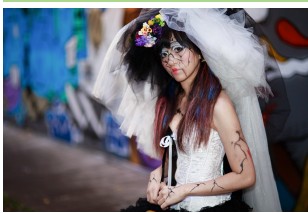 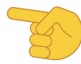

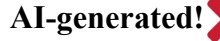 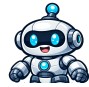

**OmniVL-Guard:<think>** Got it, let's try to figure out if this image is real, AI-generated, or tampered. First, look at the details. The person's face has some weird makeup—like the eyes and the lines around them, but wait, the hands: let's check the fingers. Oh right, the hands look distorted. Let's see: the fingers seem to be merged or have an unnatural shape, which is a common AI artifact. Also, the overall composition—maybe the background is a bit blurry, but the main issue is the hand details. AI often struggles with hands, making extra fingers or weird proportions. So probably AI-generated.**</think>**

**<answer>Answer: AI-generated.</answer>**

**Correct answer:** AI-generated.

## Case3-Video Forgery Detection

**Question:**Please determine if the video is Real, AI-generated, or Tampered.

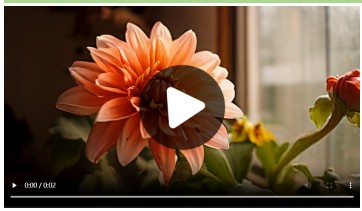 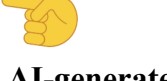 **AI-generated!** 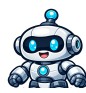

**OmniVL-Guard:<think>**Got it, let's try to figure out if this video is AI-generated. First, look at the details of the flower. The petals have a very smooth, almost too perfect texture, which is common in AI images. Then, the background—there's a window with a blurry view, but the lighting seems a bit off, not natural. Also, the movement in the video is very subtle, maybe like a slight zoom or shift, but it doesn't look like real footage. AI-generated videos often have this kind of static, almost painted look, with textures that are not quite right for real objects. So probably AI-generated.**</think>**

**<answer>Answer: AI-generated.</answer>**

**Correct answer:AI-generated.**

## Case4-Text-Image Forgery Detection

**Question:**Please determine if the news is Real, AI-generated, or Tampered.

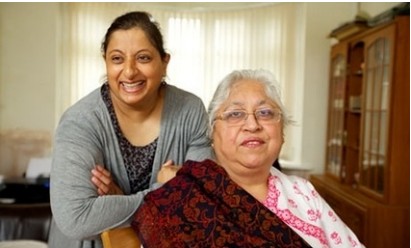

*Caption:Shazia Hamid left and Rekha Mehra discovered tumours by selfchecking.*

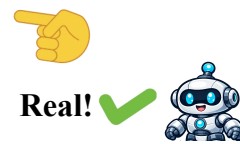

**Real!** ✅

**OmniVL-Guard:\<think\>**Got it, let's try to figure out if this is real, AI-generated, or tampered. First, look at the image details: the two women have natural facial features—no weird distortions like mismatched eyes, odd teeth, or unnatural skin textures.

The background has a wooden cabinet and curtains that look realistic, no blurriness or inconsistent lighting. Then the caption says they discovered tumors via self-checking, which is a plausible real news story (since self-detection is common). The system's initial note says high probability real, and there's no obvious sign of tampering (like edited parts that don't fit). So all signs point to real.**\</think\>**

**\<answer\>Answer: Real.\</answer\>**

**Correct answer:Real.**

## Case5-Text Tampering Location

**Question:**Please locate the tampered words in the caption and provide the indices of the altered words as a list (starting from 0).

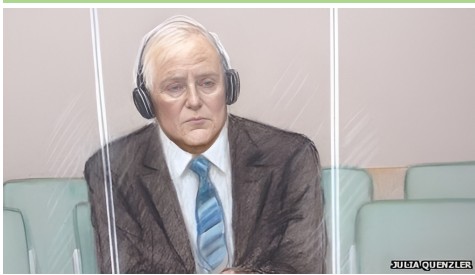

Caption:***Isabel Diaz Ayuso*** *sat in the dock as verdicts were returned.*

**OmniVL-Guard:<think>**Alright, looking at this image, it's clearly a courtroom scene with a man in a suit and headphones, probably a defendant. The caption says "Isabel Diaz Ayuso sat in the dock..." Hmm, that looks odd. Isabel is a female name, but the person here is unmistakably male. Let's check the caption against the image again. The mismatch is right at the start—the first three words, "Isabel," "Diaz," and "Ayuso," don't match the visual subject at all. This confirms the text has been altered to misidentify the person. The inconsistency is at [0, 1, 2].**</think>**

**<answer>**Tampered words list: [0, 1, 2]**</answer>**

**Correct answer:[0, 1, 2].**

## Case6-Image Tampering Location

**Question:**Please locate the tampered region in the image.

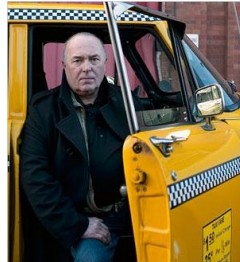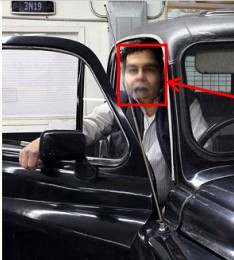

*Caption:Al Gallego and Ray Tomkinson, two taxi drivers, sit in their respective taxis, ready to take passengers on a ride.*

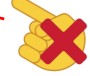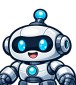

**OmniVL-Guard:<think>**Alright, looking at this image of two taxi drivers, the caption says it's Al Gallego and Ray Tomkinson. Hmm, that's odd right away—Ray Tomkinson is a footballer, not a taxi driver. That mismatch makes me suspicious the image might be altered.

Let's check the two people. The driver on the left looks fine, lighting and texture seem consistent. But the guy on the right... something feels off about his face. Wait a moment, the lighting on his face doesn't match the light coming from the taxi window; it's like the face was taken from a different photo. And the texture around his mouth and chin looks too smooth, almost airbrushed, compared to the rest of his skin. Is that just a filter? No, because when I look closer at the edges, especially around the hairline and jaw, there's a faint halo, a classic sign of a bad paste job. This confirms the face area has been manipulated. So, the tampered region is [753, 218, 840, 376].**</think>**

**<answer>Tampered bbox: {"boxes": [753, 218, 840, 376]}</answer>**

**Correct answer:[753, 218, 840, 376].**

## Case7-Video Tampering Location

**Question:**Please identify the specific time segments that have been tampered with.

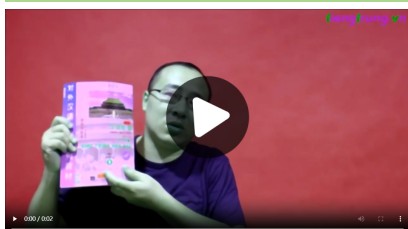

[0.5s-1.0s]

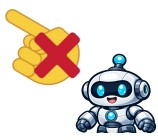

**OmniVL-Guard:<think>**Alright, let's watch this video. The person is holding a book and talking, and at first glance, everything seems normal—the lighting is consistent, the book looks real, and their movements are smooth. But wait, right around the half-second mark, something catches my eye. The book's cover starts to flicker. Hmm, that looks odd. Let's check the frames more closely. Between 0.5 and 1.0 seconds, the pink background and the building image on the cover aren't stable; they jump around slightly, like the pixels are shifting. Is that just a compression artifact? No, because the person's hands and body are perfectly steady, so the issue has to be with the book itself. This confirms it's not a camera shake. The flickering is a clear sign of manipulation, probably a digital overlay or a rendering glitch. So, the fake segment is [0.5s-1.0s].**</think>**

**<answer>Tampered segment: [0.5s-1.0s]</answer>**

**Correct answer:[0.5s-1.0s].**

