# OpenReview forum: "OmniVL-Guard: Towards Unified Vision-Language Forgery Detection and Grounding via Balanced RL"
_ICML.cc/2026/Conference — ICML 2026 regular_

### Official Review · Reviewer_Cz9n · 2026-03-08

**Soundness:** 2
**Presentation:** 3
**Significance:** 2
**Originality:** 3
**Overall Recommendation:** 4
**Confidence:** 4

**Summary:**

The paper proposes OmniVL-Guard, a unified framework for multi-modal forgery detection and grounding across text, images, and videos. It identifies task-difficulty imbalance in multi-task learning, where easier classification dominates harder localization tasks. To address this, the authors introduce a Self-Evolving CoT Generation pipeline for high-quality reasoning data and ARSPO, a reinforcement learning strategy with adaptive reward shaping and dynamic task balancing. Experiments show strong improvements over prior methods on both in-domain and zero-shot out-of-domain benchmarks, particularly for fine-grained localization tasks.

**Compliance With Llm Reviewing Policy:**

Affirmed.

**Key Questions For Authors:**

1. How sensitive is ARSPO to reward scaling hyperparameters across different datasets?
2. Does Self-Evolving CoT introduce systematic reasoning bias compared to human annotations?
3. How does the method perform when one or more modalities are missing at inference?
4. What is the computational cost compared to strong SFT-only baselines?

**Limitations:**

1. High training complexity and compute requirements due to multi-stage RL.
2. Dependence on curated datasets may limit robustness to unseen manipulation types.
3. Synthetic CoT generation may propagate model hallucinations.

**Strengths And Weaknesses:**

**Strengths**
1. Identifies difficulty imbalance in multi-task RL.
2. ARSPO is technically well-motivated with theoretical analysis.
3. Dataset (FSFR) with structured CoT supervision.

**Weakness**
1. Framework complexity is high with many interacting components.
2. Dataset construction pipeline may introduce bias from synthetic CoT.
3. Improvements on binary classification are relatively modest.

---

> ### Author Rebuttal · Authors · 2026-03-31
>
> # Response to Reviewer Cz9n
>
> We thank the reviewer for recognizing **task-difficulty imbalance**, the motivation and theory of **ARSPO**, the value of **FSFR** with **structured CoT**, and OmniVL-Guard’s strong performance. Regarding the concerns, we provide the following clarifications:
>
> > ### **Q1: Framework complexity**
>
> Our framework is indeed relatively complex, but this complexity serves the goal of a **unified foundation model**, not unnecessary design. The **Self-Evolving CoT pipeline** is introduced to build high-quality forensic reasoning data while avoiding hindsight bias and reducing difficulty bias in the cold-start set, whereas **ARSPO** addresses optimization imbalance in unified multi-task learning, especially the dominance of easier classification over harder grounding tasks.
>
> Importantly, this cost is mainly **one-time** during base-model construction. We will release the **FSFR dataset** and **training checkpoints**, so future users need not reproduce the full framework and can directly use **OmniVL-Guard as a base model** for downstream fine-tuning or adaptation.
>
> In future work, we will further simplify and streamline the framework toward a more resource-efficient design.
>
> > ### **Q2: Potential bias(hallucinations) from synthetic CoT**
>
> We agree bias(hallucinations) is an important concern, and our pipeline explicitly minimizes such bias.
>
> Synthetic CoT is used only in the **seed** and **Hard-CoT** portions, which together account for **less than 25\%** of the full CoT dataset. Thus, most of $FSFR_{sft}$ does not rely on synthetic CoT, limiting its effect on the final training distribution.
>
> Moreover, as detailed in **Appendix H**, we apply strict **dual-MLLM filtering and verification**, together with **manual spot checking**, before data inclusion. These steps are designed to reduce low-quality, noisy, hallucinated, or biased reasoning traces.
>
> > ### **Q3: Modest improvements on binary classification**
>
> In the **in-domain** setting, binary classification is already strong, so further gains are naturally limited. Under **OOD**, however, our method still shows clear binary-classification improvements, as reported in **Table 3**.
>
> More importantly, **ARSPO** is designed to address **optimization imbalance** in unified multi-task learning, where easier **binary classification** tends to dominate and hinder harder tasks such as fine-grained localization. Its main goal is therefore to improve these harder tasks, rather than maximize gains on an already strong classification branch alone.
>
> We agree that further improving binary classification remains important, and will continue exploring this direction while preserving the gains on harder grounding tasks.
>
> > ### **Q4: Sensitivity of ARSPO hyperparameters across datasets**
>
> Our method does not appear highly sensitive to reward-scaling hyperparameters. In particular, the reward slope $a$ is fixed to 3 in our main experiments and performs stably; tests with $a=1\!-\!9$ also show relatively stable results.
>
> Moreover, our setting is not to tune ARSPO separately for many independent datasets. OmniVL-Guard trains a **single unified model** for multiple forgery tasks and modality combinations. Accordingly, **FSFR** merges many single-task/single-modality datasets into one unified corpus, and the **same ARSPO hyperparameters are kept fixed throughout training**.
>
> In this sense, ARSPO is **not dataset-specific** in our main setting; its hyperparameters are shared across the diverse data sources within FSFR.
>
> > ### **Q5: Potential reasoning bias**
>
> Self-Evolving CoT shows almost no observable systematic bias relative to human annotations. As reported in Appendix **H.3**, manual quality checks give an average pass rate of about **98\%**, indicating high consistency with human judgment.
>
> > ### **Q6: Performance under missing modalities at inference**
>
> We provide two pieces of evidence. First, in **OOD**, OmniVL-Guard directly handles the unseen **text-video** modality combination, suggesting robustness to changed modality compositions.
>
> Second, we conduct an **in-domain missing-modality test** on **text-image**(DGM4) by dropping one modality at inference. The results are:
> |Setting|ACC|
> |-|-:|
> |text+image|72.3|
> |Text only|71.8|
> |Image only|72.0|
>
> These results show that OmniVL-Guard remains functional even when one modality is missing at inference.
>
> > ### **Q7: Computational cost**
>
> At inference, OmniVL-Guard has the **same computational cost** as strong **SFT-only baselines** with the same backbone size (e.g., 8B).
>
> At training, our method is more expensive than SFT-only training because it includes an additional RL stage. However, as discussed in **Q1**, this extra cost is mainly incurred during strong **base-model construction**.
>
> > ### **Q8: Robustness to unseen types**
>
> Our OOD results (Table 3) suggest that OmniVL-Guard is robust to unseen manipulation types. This is especially evident when generalizing to a new modality combination, Video+Text.

---

> > ### Author Rebuttal · Reviewer_Cz9n · 2026-04-05
> >
> > My concerns have been adequately addressed.

---

### Official Review · Reviewer_uymn · 2026-03-12

**Soundness:** 3
**Presentation:** 3
**Significance:** 4
**Originality:** 4
**Overall Recommendation:** 5
**Confidence:** 4

**Summary:**

This paper presents a unified vision-language framework and introduces FSFR, a comprehensive multimodal benchmark leveraging Chain-of-Thought (CoT) reasoning for forgery detection and grounding. To enhance the model’s forensic reasoning capabilities, the authors innovatively integrate Reinforcement Learning (RL) into the training pipeline. Furthermore, to address the learning imbalance between low-difficulty tasks (classification) and high-difficulty tasks (localization), they propose ARSPO, an adaptive reward shaping algorithm. By dynamically adjusting the gradient signals, ARSPO successfully resolves multi-task exploration bottlenecks and significantly boosts performance on challenging tasks.

**Compliance With Llm Reviewing Policy:**

Affirmed.

**Final Justification:**

I appreciate the authors’ detailed and constructive rebuttal, which convincingly resolves the issues I previously raised. In light of these clarifications, I am pleased to update my evaluation and increase my rating to accept (5).

**Key Questions For Authors:**

Questions:
1- Since difficulty bias is a general issue in multi-task RL, I suggest testing ARSPO on another standard multimodal task (e.g., joint Visual Question Answering and Dense Region Captioning) to prove its generalizability beyond forgery detection.
2- Given the instability risks of exponential growth in RL, did you experiment with safer convex mathematical functions (e.g., polynomial mappings like x^n instead of g(x)=e^(ax)) for gradient amplification?
3- Without explicit video training, how does the model handle long-form videos with subtle temporal manipulations? Does this zero-shot approach lead to context loss in extended video sequences?

**Limitations:**

I recommend adding a Limitations section discussing the heavy computational cost of the Self-Evolving CoT pipeline and the reliance on proprietary MLLMs for seed data.

**Strengths And Weaknesses:**

Strengths:
1- Novel Benchmark & Self-Evolving CoT: Introduces the comprehensive FSFR benchmark and utilizes a Self-Evolving Chain-of-Thought (CoT) approach, replacing black-box detection with interpretable, step-by-step reasoning.
2- Identifying Difficulty Bias: Accurately identifies and theoretically justifies the “difficulty bias” in multi-task learning, explaining why models tend to favor easy classification over hard localization.
3- ARSPO for Hard Tasks: Proposes the ARSPO algorithm with an exponential reward function to amplify gradient signals for high-difficulty tasks, effectively solving the multi-task exploration bottleneck.
4- Extensive Experiments & OOD Robustness: Conducts comprehensive evaluations against various state-of-the-art models, demonstrating exceptional Zero-Shot Out-of-Domain (OOD) generalization on entirely unseen data.

Weaknesses:
1- High Computational Overhead: The complex four-stage CoT generation and iterative RL pipeline is highly resource-intensive, potentially limiting reproducibility for researchers with constrained compute.
2- Reward Function Sensitivity: The exponential reward mapping (g_k(x)=e^(ax)) in ARSPO is highly sensitive to hyperparameters; slight deviations could lead to unstable RL training or gradient explosion.
3- Limited Temporal Depth: While excluding video data during training successfully proves zero-shot OOD capabilities, it may leave the model relatively underequipped for complex, long-form temporal video grounding.

---

> ### Author Rebuttal · Authors · 2026-03-31
>
> # Response to Reviewer uymn
>
> We sincerely thank the reviewer for the thoughtful feedback and for recognizing the **novel FSFR benchmark** and our **self-evolving CoT pipeline** for interpretable multimodal forgery reasoning. We also appreciate the reviewer’s recognition of our identification of **difficulty bias** in unified multi-task learning, the effectiveness of **ARSPO** in improving hard-task optimization, and our **extensive experimental validation** with strong **zero-shot OOD generalization** on unseen data. Regarding the concerns, we provide the following clarifications:
>
> > ### **Q1: Computational overhead and reproducibility of the CoT/RL pipeline**
>
> We agree that the full pipeline is computationally intensive. However, each step is carefully designed to achieve our goal: build a **unified foundation model**. In particular, Forensic Reasoning Seed Priming is used to construct a high-quality initial seed set for cold-start; Self-Evolution with iterative RL progressively expands this seed set to reduce hindsight bias and improve coverage beyond easy samples; and *Collaborative Hard-CoT Synthesis specifically supplements long-tail hard cases that remain challenging after self-evolution. Together, these stages are critical to build reliable forensic CoT data rather than an unnecessarily complex pipeline.
>
> Importantly, the cost is mainly incurred **once during base-model construction**, rather than by every future user. We will fully release the **FSFR dataset** together with the **training checkpoints**, so later researchers do not need to reproduce the full CoT generation or RL pipeline. Instead, they can directly use **OmniVL-Guard as a base model** and further fine-tune or adapt it for their own downstream settings.
>
> In future work, we will focus on distilling these components into a more streamlined and resource-efficient framework.
>
> > ### **Q2: Sensitivity of the exponential reward mapping hyperparameter**
>
> The performance variation in **Fig. 4(a)(b)** is indeed visible, but it should be interpreted together with the fact that the tested range of $a$ is deliberately large, spanning from **1 to 9**. Even under such a wide interval, the variation remains moderate (less than 2\%), which suggests that the method is not excessively fragile. In addition, we did **not** observe unstable RL training or gradient explosion in practice.
>
> Actually, the exponential reward mapping does not indiscriminately enlarge all gradients. Its role is to **amplify the optimization signal of better responses** while suppressing relatively weak ones. Thus, changing $a$ mainly adjusts the sharpness of reward discrimination, rather than causing uncontrolled growth of the overall update scale.
>
> Therefore, while $a$ should be chosen reasonably, our results suggest that ARSPO is practically stable, and **$a=3$** already serves as a reliable default.
>
> > ### **Q3: Temporal depth and long-form video grounding**
>
> Our training setup does include **video data**. What is intentionally excluded is only the **video-text pair setting** for OOD evaluation, in order to test zero-shot generalization to an unseen joint modality combination, rather than to remove video understanding from training.
>
> Since existing long-form video grounding benchmarks are still very limited, we additionally construct a small evaluation set by selecting **100 samples from FakeTT** and applying partial-segment manipulations for temporal localization testing. Below are the long-duration (**nearly 1 min**) video grounding results:
>
> |Method|Video Loc.(tIoU)|
> |-|-:|
> |Qwen3VL-235B|18.2|
> |OmniVL-Guard|56.4|
>
> > ### **Q4: Generalizability of ARSPO beyond forgery detection**
>
> ARSPO is not limited to forgery detection. We have also applied it to a **remote sensing disaster understanding** problem, which includes three task types: **classification**, **VQA-style reasoning**, and **dense region grounding/localization**. The results also support the generalizability of ARSPO beyond the forgery domain.
>
> |Method|Cls.|VQA|Dense Loc.|
> |-|-:|-:|-:|
> |SFT|72.4|42.9|39.7|
> |SFT+ARSPO|90.8|72.6|54.3|
>
> We will include these results and release the relevant ARSPO details for this remote sensing setting in the revision.
>
>
> > ### **Q5: Alternative convex reward mappings beyond exponential form**
>
> We also experimented with a polynomial convex mapping, \(g(x)=x^n\), with \(n=2.7\), and compared it with the exponential form used in the paper.
> |Reward Mapping|Bin. Cls|Image Loc.|Text Loc.|Video Loc.|
> |-|-:|-:|-:|-:|
> |$g(x)=x^{2.7}$|90.73|54.08|63.51|59.94|
> |$g(x)=e^{3x}$|90.85|54.26|63.78|59.22|
>
> The performance gap between the two mappings is small. This is also consistent with our analysis in **Sec. 4.1**: the key factor is the use of a **convex reward mapping** to amplify better responses, rather than the exact functional form itself.

---

> > ### Author Rebuttal · Reviewer_uymn · 2026-04-03
> >
> > Thank you for your response; it clearly addresses and resolves my previous concerns.

---

### Official Review · Reviewer_KEeV · 2026-03-12

**Soundness:** 3
**Presentation:** 3
**Significance:** 3
**Originality:** 3
**Overall Recommendation:** 4
**Confidence:** 3

**Summary:**

OmniVL-Guard is a unified RL framework for detection of forgery, and has grounding across multiple modalities. The paper proposes a self-evolving CoT generation that constructs high quality forensic data in an incremental manner. It also focuses of ARSPO, which uses nonlinear reward mapping and coefficient adjustment for balanced multi-task optimization. The framework is built upon Qwen3VL-8B and evaluated on a set of diverse datasets. Results show the model achieves a strong in-domain performance and also fares well in zero-shot OOD generalization.

**Compliance With Llm Reviewing Policy:**

Affirmed.

**Final Justification:**

My concerns have been fully addressed.

**Key Questions For Authors:**

See above.

**Limitations:**

yes

**Strengths And Weaknesses:**

Strengths:

1. Well-formed motivation and problem formulation using the existing literature gaps especially in Table1a highlights the need for a better forgery detection solution spanning across different modalities.
2. Structured CoT pipeline addresses the problems associated with hindsight bias.
3. Thorough evaluations and ablation study that isolates the contribution of the proposed components like SAPO, TBRNF and DCA.

Weaknesses:

1. The motivation and understanding of the dynamic coefficient algorithm seemed a bit unclear to me. There seems to be no analysis of the proposed hyperparameters and justification of their choices which makes the quantification of ARSPO's gains as robust a bit unclear. The paper might benefit from a more detailed discussion of these hyperparameters.
2. The ARSPO term is constantly confused by the authors with scaling instead of shaping, which are distinct concepts across the paper.

---

> ### Author Rebuttal · Authors · 2026-03-31
>
> # Response to Reviewer KEeV
>
> We sincerely thank the reviewer for the thoughtful feedback and for recognizing our **well-motivated problem formulation** for unified multimodal forgery analysis, the value of our **structured self-evolving CoT pipeline** for constructing high-quality forensic data, and our **thorough evaluations and ablation studies** that isolate the contributions of key components such as **SAPO, TBRMF, and DCA**. We also appreciate the reviewer’s recognition of OmniVL-Guard’s **strong in-domain performance** and **promising zero-shot OOD generalization** across diverse datasets. Regarding the concerns, we provide the following clarifications:
>
> > ### **Q1: Hyperparameter discussion of the dynamic coefficient algorithm**
>
> We agree that the motivation and role of the dynamic coefficient hyperparameters should be clarified more explicitly. In fact, these hyperparameters have different functions. $T_{warm}$ and $T$ are used to ensure that task metrics are collected over a sufficiently stable window, where $T_{warm}$ defines the baseline collection stage and $T$ controls the coefficient update frequency. $\alpha_{boost}$ and $\alpha_{decay}$ determine the magnitude of coefficient adjustment once a task is identified as lagging or already saturated. By contrast, the threshold terms are tied to the task metrics observed during warm-up, and are therefore more data-dependent.
>
> Thus, the hyperparameters that mainly control the algorithm behavior are the first four in **Table 10**. We will add sensitivity results by varying each of them by $\pm20\%$ under the full training setup. **Each result reports the average over all tasks.** The performance remains stable overall:
>
> |Param|-20%|Default|+20%|
> |-|-:|-:|-:|
> |$T_{warm}$|66.6|67.0|66.8|
> |$T$|66.8|67.0|66.7|
> |$\alpha_{boost}$|66.8|67.0|66.9|
> |$\alpha_{decay}$|66.4|67.0|66.8|
>
> For the threshold-related hyperparameters, since they are defined from warm-up task statistics, we will report a separate sensitivity study on a **10k RL subset**, again using $\pm20\%$ perturbations. **Each result again reports the average over all tasks.** The results are also stable:
>
> |Threshold group|-20%|Default|+20%|
> |-|-:|-:|-:|
> |$\epsilon_{mom},\epsilon_{rescue},\tau_{high}$|61.1|61.6|61.3|
>
> Overall, these results suggest that ARSPO is not highly fragile to moderate hyperparameter variation. We will add these clarifications and sensitivity results in the revision to make the role of the dynamic coefficient algorithm more transparent.
>
> > ### **Q2: Clarification of “scaling” vs. “shaping” in ARSPO**
>
> The reviewer is correct that **reward scaling** and **reward shaping** are distinct concepts, and our current wording may blur this distinction. We will make the terminology more precise throughout the paper, and consistently describe the exponential component as **reward shaping/remapping** rather than scaling.
>
> More precisely, in ARSPO, the **dynamic task coefficients** perform **task-level scaling**, since they adjust the optimization strength assigned to different tasks. By contrast, the **exponential reward transformation** should be described as **reward shaping/remapping**, rather than scaling, because it changes the reward sensitivity in a nonlinear manner instead of applying a constant multiplicative factor.
>
> Importantly, these two components are both introduced in the context of **multi-task optimization**. The dynamic scaling component explicitly balances optimization across tasks, while the reward shaping component further improves the relative optimization signal under the multi-task setting by enlarging the distinction between better and worse responses. In other words, ARSPO is not merely a scaling strategy, but a combination of **task-aware scaling** and **reward shaping** designed for unified multi-task training.

---

> > ### Author Rebuttal · Reviewer_KEeV · 2026-04-01
> >
> > Thanks, I decide to maintain my score.

---

### Official Review · Reviewer_d8hU · 2026-03-13

**Soundness:** 3
**Presentation:** 3
**Significance:** 4
**Originality:** 3
**Overall Recommendation:** 4
**Confidence:** 3

**Summary:**

This paper proposes **OmniVL-Guard**, aiming to address multimodal forgery detection and manipulation localization within a unified framework. The authors argue that in a multi-task setting simple supervised fine-tuning may be dominated by the easier classification task, which can hurt fine-grained localization performance. To address itt, the paper introduces a reinforcement learning–based method ARSPO to dynamically balance the optimization across different tasks.

**Compliance With Llm Reviewing Policy:**

Affirmed.

**Final Justification:**

I tend to keep my original score.

**Key Questions For Authors:**

i) The paper emphasizes that the method targets omnibus vision–language forgery detection and grounding. However, the main text states that video–text pairs are intentionally excluded from the training set and are only used for OOD zero-shot testing. This means that while the evaluation covers more modality combinations, the model is not actually trained to jointly learn all cross-modal settings. It would therefore be helpful if the authors could clarify this design choice; the claim of a “unified” framework might need to be phrased more cautiously.

ii) I am also curious how strongly Self-Evolving CoT Generation depends on closed-source or highly proprietary models. If weaker or open-source teacher models are used instead, would the pipeline still work effectively? The current CoT cold-start process relies on several strong MLLMs (e.g., Seed, Gemini, ChatGPT) for generation, verification, and refinement, and Hard-CoT synthesis further adopts multi-agent collaboration. While comprehensive, this design may make the final performance heavily dependent on the quality of external models.

iii) The analysis in the paper attributes the performance gain to difficulty bias in multi-task training, where classification dominates optimization, and proposes ARSPO with dynamic coefficients and nonlinear reward mapping to address this issue. However, the ablation and single-task results show that the exponential reward mapping alone already provides noticeable improvements, even without multi-task competition. This raises the question of whether the gains from ARSPO mainly come from balancing multi-task optimization, or largely from the more aggressive reward remapping itself. Clarifying this point would help better understand the source of improvement.

**Limitations:**

Yes,

**Strengths And Weaknesses:**

Strengths

One strength of the paper is that it targets a practical yet relatively fragmented problem: handling text, image, and video within a unified framework for both forgery detection and manipulation localization, rather than focusing only on single-modality or bimodal settings. This unified perspective is, in my view, a meaningful direction for real-world applications.

Weaknesses

i) The CoT cold-start and data construction pipeline relies on several SOTA MLLMs (e.g., Seed1.6, Gemini, ChatGPT-5) for generation, verification, and refinement, and the Hard-CoT synthesis even requires multi-agent collaboration. While this pipeline is engineering-wise complete, it may introduce practical challenges in terms of training cost, API accessibility, and reproducibility.

ii) In the OOD evaluation, the paper mainly reports binary classification results without further assessing the localization task. Moreover, many MLLMs are evaluated in a zero-shot setting, while the proposed method and some domain-specific baselines are trained on FSFR before testing. This comparison therefore resembles “general models vs. domain-trained models,” which may not fully isolate the benefit brought by the proposed method itself. In other words, while the results suggest strong performance, it remains unclear how much of the gain can be attributed specifically to the RL/ARSPO component without stricter controlled comparisons.

---

> ### Author Rebuttal · Authors · 2026-03-31
>
> # Response to Reviewer d8hU
>
> We thank the reviewer for recognizing the value of our unified framework for **text, image, and video** forgery **detection** and **localization**. We address the concerns below:
>
> > ### **Q1: Practicality and reproducibility of the pipeline**
>
> Our pipeline is not unnecessarily complex; each stage addresses a specific CoT construction challenge towards a **unified foundation model**.
>
> Multiple MLLMs are used to avoid **hindsight bias**. Our preliminary results show that directly injecting GT into CoT generation causes annotation-oriented shortcuts rather than genuine forensic reasoning. We therefore adopt several MLLMs in our self-evolving forensic CoT to remedy this issue.
>
> Moreover, self-evolution still struggles on **long-tail hard samples**, which may bias data toward easier cases. We thus introduce collaborative Hard-CoT synthesis with multi-agent to supplement these samples and broaden the difficulty spectrum. This is to reduce difficulty bias in cold-start data.
>
> We also emphasize that this cost is **one-time**: we will release the CoT data and training checkpoints, so future users can directly use **OmniVL-Guard as a base model** for lightweight fine-tuning without reproducing the full pipeline.
>
> > ### **Q2: OOD evaluation on localization and controlled comparison of method gains**
>
> For localization, suitable **OOD benchmarks with fine-grained forgery grounding** are still very limited, and the few available ones (e.g., ForgeryNet, DGM4) have already been included in our training setup. We therefore could not identify a fair benchmark for OOD localization evaluation. We agree that more challenging OOD grounding benchmarks would be valuable, and will continue exploring this direction in future work.
>
> For the second concern, we add a controlled comparison using the **same Qwen3VL-8B backbone** trained on **FSFR with SFT only**, and compare its **average OOD accuracy** with **OmniVL-Guard**, thereby separating the effect of domain training from the additional gain of **RL/ARSPO**:
> |Method|Avg. OOD ACC|
> |-|-:|
> |Qwen3VL-235B|67.06|
> |Qwen3VL-8B (SFT)|57.13|
> |OmniVL-Guard|75.07|
>
> > ### **Q3: Clarification of the “unified” claim under missing video-text training**
>
> We exclude **video-text pairs** from training because existing benchmarks are limited and highly homogeneous, mostly from **Douyin/TikTok**. Including them would weaken OOD fairness, since the test distribution would substantially overlap with the training source.
>
> In our paper, **omnibus/unified** means that a **single framework** handles multiple vision-language forgery input formats within one model, rather than using separate models for different modality combinations. Achieveing this goal does not require training on every combination. As supported by both in-domain and OOD results, OmniVL-Guard is capable of processing diverse multimodal inputs even video-text training data is missing.
>
> We will use **“unified”** more carefully in the revision, and continue exploring more diverse video-text benchmarks in future work.
>
> > ### **Q4: Dependence of Self-Evolving CoT Generation on closed-source teachers**
>
> The main gain comes from the **pipeline design**, not any specific teacher. To test dependence on closed-source models, we replace the original pool with three open-source MLLMs: **GLM-4.6V-106B, Qwen3VL-235B, and InternVL3.5-241B**. Due to time limits, we construct a **6k CoT subset** and compare: (1) Direct GT, (2) open-source teachers+Self-Evolving CoT, and (3) the standard setting.
>
> |Method|Bin.Cls.|Img.Loc.|Text.Loc.|Vid.Loc.|
> |-|-:|-:|-:|-:|
> |Direct GT(6k)|77.6|46.7|45.9|42.3|
> |Open-source teachers+Self-Evolving(6k)|83.5|48.4|52.8|48.7|
> |Standard(6k)|85.1|50.0|55.2|50.1|
>
> These results show that **Self-Evolving CoT** does not depend critically on closed-source teachers: even with open-source teachers only, it still clearly outperforms **Direct GT** under the same 6k budget, while the standard setting is only slightly better.
>
> > ### **Q5: Source of ARSPO gains beyond multi-task difficulty bias**
>
> ARSPO gains come from both **multi-task balancing** and **reward remapping**; the two are complementary and both are core to our method.
>
> First, in our unified setting, task difficulty differs, so **task-specific reward mapping** is itself part of balancing. ARSPO balances tasks not only with dynamic coefficients, but also by using sharper reward shaping for harder localization tasks.
>
> Second, ARSPO also works in single-task settings because nonlinear reward remapping has a more general effect: it amplifies better responses and improves gradient discrimination.
>
> Thus, our point is not that ARSPO works only because of multi-task difficulty bias. Rather, **difficulty bias makes such reward shaping especially necessary** in the unified setting, while nonlinear reward remapping is itself another key source of improvement.

---

> > ### Author Rebuttal · Reviewer_d8hU · 2026-04-02
> >
> > Thanks for the clarification. Based on that, I have decided to keep the current score.

---

### Decision · Program_Chairs · 2026-04-30

**Decision:**

Accept (regular)

**Comment:**

This paper proposes OmniVL-Guard, a unified framework for multimodal (text, image, video) forgery detection and grounding, combining a self-evolving CoT data construction pipeline with an RL-based optimization method (ARSPO) for balancing multi-task learning.

All reviewers acknowledge the importance of the problem and the strength of the empirical results, particularly in improving fine-grained localization and demonstrating promising zero-shot generalization. The proposed framework is well-motivated and integrates multiple components (data construction and RL optimization) into a coherent system.

The main point of discussion concerns the source of the performance gains and the clarity of the contribution. Reviewer concerns that the improvements may largely stem from reward shaping rather than the proposed multi-task balancing mechanism are valid. While the rebuttal clarifies that ARSPO combines both reward shaping and dynamic task weighting, the current empirical evidence does not fully disentangle these factors. In addition, the evaluation setup—particularly in OOD comparisons—mixes domain-trained models with zero-shot baselines, making it harder to precisely isolate the contribution of the proposed method.

Concerns regarding the complexity of the CoT data pipeline and reliance on strong external models are noted but are partially mitigated by the authors’ plan to release datasets and checkpoints. The “unified” claim is somewhat overstated, as not all modality combinations are seen during training, but this does not substantially undermine the overall contribution.

Overall, this paper presents a solid and practically relevant system with consistent empirical improvements, despite some limitations in attribution and evaluation rigor. The contribution is incremental but meaningful for the multimodal forgery detection setting.